# On the difficulty of learning chaotic dynamics with RNNs

**Jonas M. Mikhaeil**[1,2,*], **Zahra Monfared**[1,4*], and **Daniel Durstewitz**[1,2,3]

j.mikhaeil@columbia.edu, {zahra.monfared, daniel.durstewitz}@zi-mannheim.de

[1]Department of Theoretical Neuroscience, Central Institute of Mental Health, Medical Faculty Mannheim, Heidelberg University, Mannheim, Germany
[2]Faculty of Physics and Astronomy, Heidelberg University, Heidelberg, Germany
[3]Interdisciplinary Center for Scientific Computing, Heidelberg University
[4]Department of Mathematics & Informatics and Cluster of Excellence STRUCTURES, Heidelberg University, Heidelberg, Germany
[*]These authors contributed equally

## Abstract

Recurrent neural networks (RNNs) are wide-spread machine learning tools for modeling sequential and time series data. They are notoriously hard to train because their loss gradients backpropagated in time tend to saturate or diverge during training. This is known as the exploding and vanishing gradient problem. Previous solutions to this issue either built on rather complicated, purpose-engineered architectures with gated memory buffers, or - more recently - imposed constraints that ensure convergence to a fixed point or restrict (the eigenspectrum of) the recurrence matrix. Such constraints, however, convey severe limitations on the expressivity of the RNN. Essential intrinsic dynamics such as multistability or chaos are disabled. This is inherently at disaccord with the chaotic nature of many, if not most, time series encountered in nature and society. It is particularly problematic in scientific applications where one aims to reconstruct the underlying dynamical system. Here we offer a comprehensive theoretical treatment of this problem by relating the loss gradients during RNN training to the Lyapunov spectrum of RNN-generated orbits. We mathematically prove that RNNs producing stable equilibrium or cyclic behavior have bounded gradients, whereas the gradients of RNNs with chaotic dynamics always diverge. Based on these analyses and insights we suggest ways of how to optimize the training process on chaotic data according to the system's Lyapunov spectrum, regardless of the employed RNN architecture.

## 1 Introduction

Recurrent neural networks (RNNs) are widely used across various fields in engineering and science for learning sequential tasks or modeling and predicting time series [55]. Yet, they struggle when long-term temporal dependencies, very slow, or hugely varying time scales are involved [6, 34, 53, 73, 80]. Time series or sequential data with such properties are, however, very common in fields like climate physics [84], neuroscience [22, 75], ecology [87], or language processing [11]. Training RNNs on such data is hard because the loss gradients backpropagated in time easily saturate or diverge in this process. This is commonly referred to as the exploding and vanishing gradient problem (EVGP) [6, 34, 67].

One solution to the EVGP is based on specifically designed RNN architectures with gating mechanisms, such as long short-term memory (LSTM) [35] or gated recurrent units (GRU) [10]. These architectures allow states at earlier time steps to more easily influence activity much later through a

36th Conference on Neural Information Processing Systems (NeurIPS 2022).

kind of protected memory buffer, thus alleviating the EVGP by structural design. In practice, such models need to be backed up by further techniques like gradient clipping to keep the gradients in check [67]. The relatively complex architectural design of these networks impedes their mathematical analysis and requires reverse engineering after training [57, 63, 64, 80]. Partly to forego these complications, a variety of other solutions has been proposed recently, imposing restrictions on the recurrence matrix to bound the gradients [4, 9], or enforcing global stability by design or regularization [19, 49]. Often these procedures dramatically curtail the expressivity of the RNN [45, 66, 80]; in particular, they rule out chaotic dynamics (for reasons discussed further below).

This is at odds with the plethora of chaotic phenomena in nature, engineering, and society. Chaotic dynamics are commonplace, almost default in any complex physical or biological system. This includes scientific areas as diverse as neuroscience [17, 93], physiology [46], geophysics [81], climate systems [88], astrophysics [52], ecology [14], chemical reactions [21], cell [65] or population [59] biology. Chaotic phenomena are also crucial for the understanding of societal and epidemiological processes, such as the spread of diseases [58] or in economics [20]. They are further relevant in purely technical contexts such as electrical engineering [41, 83] or laser optics [43]. They have even been suggested to play an up to now largely neglected, but potentially very significant role in speech recognition [77] and natural language processing [36]. Hence, in almost any practical setting, chaotic phenomena abound. They cannot, in general, be ignored when devising RNN training algorithms.

Here we offer a comprehensive theoretical treatment of the relation between RNN dynamics and the behavior of the loss gradients during training. We find a close connection between an RNN's loss gradients and the largest Lyapunov exponent of its freely generated orbits. We mathematically prove that RNNs producing stable fixed point or cyclic behavior have bounded gradients. Crucially, however, the loss gradients of RNNs producing chaotic dynamics always diverge. Hence, the chaotic nature of many time series data induces a *principle* problem, and, despite significant efforts in the past to solve the EVGP, training RNNs on such data remains an open issue. We illustrate the implications of our theory for RNN training on several simulated and empirical chaotic time series, and adapt the idea of *sparsely forced* Back-Propagation Through Time (BPTT) as a simple yet effective remedy that enables to learn the underlying dynamics despite exploding gradients.

## 2 Related works

*Exploding and vanishing gradients.* While 'classical' remedies of the EVGP [6, 34, 67] rest on purpose-tailored architectures with gating mechanisms, which safeguard information flow across longer temporal distances [10, 35], the focus has recently shifted to simpler RNNs that address the EVGP by restricting the recurrence matrix to be orthogonal [31, 32, 37], unitary [4], or antisymmetric [9], or by ensuring globally stable fixed point solutions [38, 40], for example through co-trained Lyapunov functions [49]. However, all these approaches impose strong limitations on the dynamical repertoire of the RNN, enforcing global convergence to fixed points or simple cycles.[1] In doing so, they drastically reduce the expressiveness of these models [45, 66]. To address this problem, Erichson et al. [19] somewhat relaxed the constraints on the recurrence matrix by introducing a skew-symmetric decomposition combined with a Lipschitz condition on the activation function. Another recent approach discretizes oscillator ordinary differential equations (ODEs) to arrive at a stable system of coupled [73] or independent [74] oscillators which increase the RNN's expressiveness while bounding its gradients. By design (and as acknowledged by the authors), neither of these architectures is capable of producing chaotic dynamics, however, as the underlying ODEs do not allow for exponential divergence of close-by trajectories (a prerequisite for chaos). Given these often principle limitations of parametrically or dynamically strongly constrained models, a fruitful direction may be to modify the training process itself, e.g. through modified or auxiliary loss functions [80, 85], or special procedures for parameter updating [39] or loss truncation [61, 95]. Our empirical evaluation will follow up on such ideas, but also highlight that simple loss truncation, windowing, or architectural solutions like LSTMs are not sufficient.

*Learning dynamical systems.* Surprisingly disconnected from the work on the EVGP and learning long-term dependencies, a huge and long-standing literature deals with training RNNs on nonlinear dynamical systems (DS) [70, 86, 89], including chaotic systems like the famous Lorenz equations [56] or chaotic turbulence in fluid dynamics [54, 69]. Of those, methods based on reservoir computing [69] are special in that they start with a large complex dynamics-enabling repertoire to begin with

---

[1] We make this point more formal in Appx. A.1.6.

for which a linear mapping onto the observations in a feedback loop with the reservoir is learned (see [8] for issues associated with this strategy in the context of DS reconstruction). Teacher forcing (TF; [13, 70, 95], see also [28]) is one of the earliest techniques introduced to keep RNN trajectories on track while training. The idea behind TF is to simply replace RNN states by observations when available, thereby also effectively cutting off the gradients. TF essentially derives from ideas in dynamical control theory, and adaptive schemes that increasingly hand over control to the RNN throughout training have been devised [1, 2, 5]. A related technique with applications in control theory is "multiple shooting" [92]: Here the whole observed time series is chopped into chunks, and for each chunk of trajectory a new initial condition is estimated. Explicit constraints ensure continuity between the separate trajectory bits during optimization. State space models and the Expectation-Maximization algorithm became popular particularly in the 90es for uncovering the latent dynamics underlying a set of time series observations [23], and remain an important tool until today [16, 50]. Most recently, approaches based on variational inference and the reparameterization trick [48], like sequential variational autoencoders (SVAE), gained in popularity for DS approximation [33, 51]. "Deterministic" RNNs (i.e., with latent states not treated as random variables), like conventional LSTMs [90], remain top choices for DS reconstruction, however.

Although connections between DS ideas and loss gradients have been drawn early on [6], so far only particular scenarios (like fixed point attractors) have been considered. Closest to our work is recent work by Schmidt et al. [80], where non-divergence of loss gradients is established when RNNs converge to fixed points or cycles. However, this was done only for the particular class of piecewise-linear RNNs (PLRNNs), more restrictive conditions for cycles were imposed than assumed here, and - most importantly - the chaotic case on which we focus here was not considered. Recent studies [18, 91] also point out the general connections between Lyapunov exponents and loss gradients that we develop in sect. 3.1, but do not provide any in-depth theoretical treatment, proofs, or empirical evaluation of methods to alleviate exploding gradients in training, as we do here. Thus, a systematic theoretical framework that relates RNN dynamics more generally, and across a range of different RNN architectures, to the behavior of its training gradients, is still lacking so far.

## 3 Theoretical analysis: Relation between RNN dynamics and loss gradients

In our analysis, we will cover all major types of asymptotic dynamics (fixed points, cycles, chaos, and quasi-periodicity), and mathematically investigate their implications for the loss gradients. We will do this for all major classes of RNNs, including standard RNNs with largely arbitrary activation function, LSTMs, GRUs, and PLRNNs. The next section will first develop and illustrate the basic intuition behind the relations between RNN dynamics and loss gradients.

### 3.1 Preliminaries: RNN dynamics and loss gradients

Formally, all popular RNN architectures, including LSTMs, GRUs, or PLRNNs, are discrete time DS, defined by a (first-order-Markovian) recursive prescription for the temporal evolution of the latent states $z_t \in \mathbb{R}^M$ of the general form

$$z_t = F_\theta(z_{t-1}, s_t), \tag{1}$$

where $s_t \in \mathbb{R}^N$ is the input at time $t$ and $\theta$ are RNN parameters. Map $F_\theta$ may be instantiated by any of the common RNN architectures: For instance, for standard RNNs we have $F_\theta(z_{t-1}, s_t) = f(Wz_{t-1} + Bs_t + h)$, where $f$ is an element-wise activation function like $\tanh$ or a rectified linear unit (ReLU), $W$ a connection matrix, matrix $B$ weighs the inputs, and $h$ is the usual bias term (see sect. A.1.3–A.1.6 for the definition of other RNN models explored here).

Assuming we start at some initial value $z_1 \in \mathbb{R}^M$, and given a sequence of external inputs $S = \{s_t\}$, we can recursively rewrite eq. (1) as

$$z_T = F_\theta(F_\theta(F_\theta(...F_\theta(z_1, s_2)...))) =: F_\theta^{T-1}(z_1, \{s_t\}). \tag{2}$$

In DS theory, we characterize the long-term behavior of such sequences by its spectrum of Lyapunov exponents. The Lyapunov exponents estimate the exponential growth rates in different local directions of the system's state space, and the largest Lyapunov exponent gives the dominant exponential behavior. Let us denote the system's Jacobian at time $t$ by

$$J_t := \frac{\partial F_\theta(z_{t-1}, s_t)}{\partial z_{t-1}} = \frac{\partial z_t}{\partial z_{t-1}}. \tag{3}$$

Then, the maximum Lyapunov exponent along an RNN trajectory $\{z_1, z_2, \cdots, z_T, \cdots\}$ is defined as

$$\lambda_{max} := \lim_{T \to \infty} \frac{1}{T} \log \left\| \prod_{r=0}^{T-2} J_{T-r} \right\|, \tag{4}$$

where $\| \cdot \|$ denotes the spectral norm (or any subordinate norm) of a matrix. If $\lambda_{max} < 0$ nearby trajectories will ultimately converge to a fixed point or cycle, while for $\lambda_{max} > 0$ (a necessary condition for chaos) initially nearby trajectories will exponentially separate, i.e. we will have divergence along one (or more) directions in state space. This accounts for the sensitive dependence on initial conditions in chaotic systems.

Now let $\mathcal{L}(\boldsymbol{\theta})$ be some loss function employed for RNN training that decomposes in time as $\mathcal{L} = \sum_{t=1}^{T} \mathcal{L}_t$. Suppose we fancy BPTT as our training algorithm (similar derivations could be performed for Real Time Recurrent Learning [RTRL]), we recursively develop the loss gradients w.r.t. some RNN parameter $\theta$ in time (i.e., across layers of the RNN unrolled in time) as

$$\frac{\partial \mathcal{L}}{\partial \theta} = \sum_{t=1}^{T} \frac{\partial \mathcal{L}_t}{\partial \theta} \quad \text{with} \quad \frac{\partial \mathcal{L}_t}{\partial \theta} = \sum_{r=1}^{t} \frac{\partial \mathcal{L}_t}{\partial z_t} \frac{\partial z_t}{\partial z_r} \frac{\partial^+ z_r}{\partial \theta}, \tag{5}$$

and

$$\frac{\partial z_t}{\partial z_r} = \frac{\partial z_t}{\partial z_{t-1}} \frac{\partial z_{t-1}}{\partial z_{t-2}} \cdots \frac{\partial z_{r+1}}{\partial z_r}$$

$$= \prod_{k=0}^{t-r-1} \frac{\partial z_{t-k}}{\partial z_{t-k-1}} = \prod_{k=0}^{t-r-1} J_{t-k}, \tag{6}$$

where $\partial^+$ denotes the immediate derivative. Now observe that the behavior of the loss gradients crucially depends on the *product series* of Jacobians in eqn. (6): If the maximum absolute eigenvalues of the Jacobians $J_t$ will, in the geometric mean, be larger than 1 (i.e., $\left\| \prod_{r=0}^{T-2} J_{T-r} \right\|^{1/T} > 1$), gradients will explode as $T \to \infty$, while they will saturate if $\left\| \prod_{r=0}^{T-2} J_{T-r} \right\|^{1/T} < 1$. Thus, the key point to note is that the same terms that occur in the definition of the Lyapunov spectrum, eqn. (4), resurface in the loss gradients, eqn. (5) & (6). This accounts for the tight links between system dynamics and gradients.

## 3.2 Fixed points and cyclic dynamics

Let us start by considering the simplest types of limit dynamics that can occur in RNNs (or any discrete-time DS): fixed points and cycles. In fact, by far most of the literature on global stability in RNNs and on loss gradients focused on just fixed points [9, 19, 49], with only few authors who recently started to also connect cyclic behavior to loss gradients [73, 80]. Recall that a fixed point of a recursive map $z_t = F(z_{t-1})$ is defined as a point $z^*$ for which we have $z^* = F(z^*)$.[2] Likewise, a $k$-cycle ($k > 1$) is a set of temporally consecutive periodic points $P_k := \{z_{t_1}, z_{t_2}, \ldots, z_{t_k}\} = \{z_{t_1}, F(z_{t_1}), \ldots, F^{k-1}(z_{t_1})\}$ that we obtain from recursive application of the map such that each of the cyclic points $z_{t_r} \in P_k$ is a fixed point of the $k$ times iterated map $F^k$ (with $k$ being the smallest positive integer for which this holds). To simplify the subsequent treatment, we will collectively refer to fixed points and cycles as $k$-cycles ($k \geq 1$). Further recall that a fixed point or $k$-cycle is called *stable* if the maximum absolute eigenvalue of the Jacobian evaluated at that point is smaller than 1, *neutrally stable* if exactly 1, and *unstable* otherwise. Although the results we develop in this and the following sections will hold more widely, we will restrict our attention to recursive maps $F_{\boldsymbol{\theta}}$ from the class of RNNs $\mathcal{R} = \{\texttt{standardRNN}, \texttt{LSTM}, \texttt{GRU}, \texttt{PLRNN}\}$ (see Appx. A.1 for details).

Based on the observations made in the previous sections we can state the following theorem that links RNN dynamics and loss gradients:

**Theorem 1.** *Consider an RNN $F_{\boldsymbol{\theta}} \in \mathcal{R}$ parameterized by $\boldsymbol{\theta}$, and assume that it converges to a stable fixed point or $k$-cycle $\Gamma_k$ ($k \geq 1$) with $\mathcal{B}_{\Gamma_k}$ as its basin of attraction. Then for every $z_1 \in \mathcal{B}_{\Gamma_k}$ (i) the Jacobian $\frac{\partial z_T}{\partial z_1}$ exponentially vanishes as $T \to \infty$; (ii) for $\Gamma_k$ the tangent vectors $\frac{\partial z_T}{\partial \theta}$ and thus*

---

[2]From here on we will suppress the explicit dependence on external inputs $s_t$ notation-wise, see Remark 2.

the gradient of the loss function, $\frac{\partial \mathcal{L}_T}{\partial \boldsymbol{\theta}}$, will be bounded from above, i.e. will not diverge for $T \to \infty$; and (iii) for the PLRNN (27) both $\left\| \frac{\partial \boldsymbol{z}_T}{\partial \theta} \right\|$ and $\left\| \frac{\partial \mathcal{L}_T}{\partial \theta} \right\|$ will remain bounded for every $\boldsymbol{z}_1 \in \mathcal{B}_{\Gamma_k}$ as $T \to \infty$.

*Proof.* $(i)$ Assume that $\Gamma_k$ is a stable $k$-cycle ($k \geq 1$) denoted by

$$\Gamma_k = \{\boldsymbol{z}_1, \boldsymbol{z}_2, \cdots, \boldsymbol{z}_T, \cdots\} = \{\boldsymbol{z}_{t^{*k}}, \boldsymbol{z}_{t^{*k}-1}, \cdots,$$

$$\boldsymbol{z}_{t^{*k}-(k-1)}, \boldsymbol{z}_{t^{*k}}, \boldsymbol{z}_{t^{*k}-1}, \cdots, \boldsymbol{z}_{t^{*k}-(k-1)}, \cdots\}. \tag{7}$$

Then, the largest Lyapunov exponent of $\Gamma_k$ is given by

$$\lambda_{\Gamma_k} = \lim_{t \to \infty} \frac{1}{t} \ln \left\| J_t^* J_{t-1}^* \cdots J_2^* \right\|$$

$$= \lim_{j \to \infty} \frac{1}{jk} \ln \left\| \left( \prod_{s=0}^{k-1} J_{t^{*k}-s} \right)^j \right\|. \tag{8}$$

By assumption of stability of $\Gamma_k$ we have $\lambda_{\Gamma_k} < 0$ and also $\rho\left( \prod_{s=0}^{k-1} J_{t^{*k}-s} \right) < 1$ (the spectral radius), which implies

$$\lim_{t \to \infty} J_t^* J_{t-1}^* \cdots J_2^* = \lim_{j \to \infty} \left( \prod_{s=0}^{k-1} J_{t^{*k}-s} \right)^j = 0. \tag{9}$$

Now suppose that $\mathcal{O}_{\boldsymbol{z}_1}$ is an orbit of the map eqn. (1) converging to $\Gamma_k$, i.e. $\boldsymbol{z}_1 \in \mathcal{B}_{\Gamma_k}$. Since $\mathcal{O}_{\boldsymbol{z}_1}$ and $\Gamma_k$ have the same largest Lyapunov exponent, we have

$$\lambda_{\mathcal{O}_{\boldsymbol{z}_1}} = \lim_{T \to \infty} \frac{1}{T} \ln \| J_T J_{T-1} \cdots J_2 \| = \lambda_{\Gamma_k} < 0, \tag{10}$$

and hence for $\boldsymbol{z}_1 \in \mathcal{B}_{\Gamma_k}$

$$\lim_{T \to \infty} \left\| \frac{\partial \boldsymbol{z}_T}{\partial \boldsymbol{z}_1} \right\| = \lim_{T \to \infty} \| J_T J_{T-1} \cdots J_2 \| = 0. \tag{11}$$

$(ii) \& (iii)$ See Appx. A.2.1. $\qquad \square$

**Remark 1.** *The result of Theorem 1 part $(i)$ will be generally true for any first-order-Markovian recursive map (1), but the conclusions in part $(ii)$ may hinge on its specific definition.*

**Remark 2.** *None of the results above and throughout sect. 3 require the dynamics to be autonomous, the theory applies whether there is external input or not. In fact, mathematically, non-autonomous (externally forced) systems can always be rewritten as autonomous dynamical systems [3, 71, 97], see Appx. A.1.1 for details.*

The results above ensure that loss gradients will not diverge (explode) as $T \to \infty$ in RNNs that are "well-behaved" in the sense that they converge to a fixed point or cycle (see Fig. 1a). This is a generalization of the results given in Theorem 1 in Schmidt et al. [80], where this was shown only a) for the specific class of PLRNNs and b) for specific constraints imposed on the eigenvalue spectrum of the RNN's Jacobians which were relaxed in our theorem above.

While our treatment above is centered on the "exploding-gradients" case, various architectural modifications or regularization techniques can ensure that gradients do not vanish either, i.e. remain bounded from below as well. This was established, for instance, in Schmidt et al. [80] for PLRNNs using 'manifold attractor regularization'. In Appx. A.2.1 we show that the results from Theorem 2 from Schmidt et al. [80] on *doubly* bounded (from below and above) loss gradients can indeed be extended to the more general case covered by Theorem 1 above.

### 3.3 Chaotic dynamics

We will now consider the all-important chaotic case. Let $F$ be a recursive map and $\mathcal{O}_{\boldsymbol{z}_1} = \{\boldsymbol{z}_1, \boldsymbol{z}_2, \boldsymbol{z}_3, \cdots\}$ be an orbit of $F$. The orbit is chaotic if (i) it is not asymptotically periodic and (ii) has at least one positive Lyapunov exponent [26, 60]. If the system's invariant set is *bounded*, condition (ii) is considered a standard signature of chaos, as in this case two nearby orbits separate exponentially fast, but at the same time their mutual separation cannot go to infinity so that there are also folds. The following theorem states the sufficient condition for exploding gradients:

**(a)**

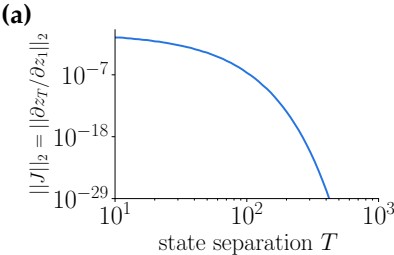

**(b)**

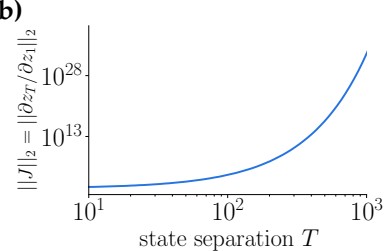

Figure 1: Illustration of the exploding gradient problem when training RNNs on dynamical systems. Jacobians (a) decay away across time separation $T$ for an RNN trained on a simple cycle (cf. Thm. 1), but (b) quickly shoot through the roof when training was performed on chaotic time series (Lorenz system; cf. Thm. 2). Note the doubly-logarithmic scale of these graphs.

**Theorem 2.** *Suppose that an RNN $F_{\boldsymbol{\theta}} \in \mathcal{R}$ (parameterized by $\boldsymbol{\theta}$) has a chaotic attractor $\Gamma^*$ with $\mathcal{B}_{\Gamma^*}$ as its basin of attraction. Then, for almost every orbit with $\boldsymbol{z}_1 \in \mathcal{B}_{\Gamma^*}$, (i) the Jacobians connecting temporally distal states $\boldsymbol{z}_T$ and $\boldsymbol{z}_t$ ($T \gg t$), $\frac{\partial \boldsymbol{z}_T}{\partial \boldsymbol{z}_t}$, will exponentially explode for $T \to \infty$, and (ii) the tangent vector $\frac{\partial \boldsymbol{z}_T}{\partial \boldsymbol{\theta}}$ and so the gradients of the loss function, $\frac{\partial \mathcal{L}_T}{\partial \boldsymbol{\theta}}$, will diverge as $T \to \infty$.*

*Proof.* Let the RNN $F_{\boldsymbol{\theta}} \in \mathcal{R}$ have a chaotic orbit denoted by $\Gamma^* = \{\boldsymbol{z}_1^*, \boldsymbol{z}_2^*, \cdots, \boldsymbol{z}_T^*, \cdots\}$. Then, denoting by $J_T^*$ the Jacobian of (1) at $\boldsymbol{z}_T^* \in \Gamma^*$, the largest Lyapunov exponent of $\Gamma^*$ is given by

$$\lambda = \lim_{T \to \infty} \frac{1}{T} \ln \left\| J_T^* J_{T-1}^* \cdots J_2^* \right\|. \tag{12}$$

Since $\Gamma^*$ is chaotic, so $\lambda > 0$. Hence, from (12), it is concluded that

$$\lim_{T \to \infty} \left\| J_T^* J_{T-1}^* \cdots J_2^* \right\| = \lim_{T \to \infty} \left\| \frac{\partial \boldsymbol{z}_T^*}{\partial \boldsymbol{z}_t^*} \right\| = \infty, \quad T \gg t. \tag{13}$$

Now, according to Oseledec's multiplicative ergodic Theorem, almost all the points in the basin of attraction of $\Gamma^*$ have the same largest Lyapunov exponent $\lambda$. Thus, (13) holds for almost every $\boldsymbol{z}_1 \in \mathcal{B}_{\Gamma^*}$.

$(ii)$ See Appx. A.2.2. $\qquad \square$

**Remark 3.** *The first part of Theorem 2 holds for all first-order-Markovian recursive maps (1). Note that for LSTMs, $\frac{\partial \boldsymbol{z}_T}{\partial \boldsymbol{z}_t}$ ($\boldsymbol{z} := (\boldsymbol{h}, \boldsymbol{c})^{\mathsf{T}}$) denotes the full Jacobian of both hidden and cell states.*

We collect some further mathematical results and remarks related to Theorem 2 in Appx. A.3.1.

Hence, the essential result is that for all popular RNNs $\mathcal{R}$ and activation functions, loss gradients will inevitably diverge if the RNN latent states converge to a chaotic attractor (as illustrated in Fig. 1b).

## 3.4 Quasi-periodicity

Quasi-periodicity is a long-term behavior which occurs on a torus and, superficially, bears some similarity to chaos in the sense that, strictly speaking, orbits are also *aperiodic*. That is, as $T \to \infty$, trajectories will never close up with themselves. Moreover, every trajectory becomes arbitrarily close to any point on the torus, that is, it is dense. One important difference between quasi-periodic and chaotic systems is, however, that in a quasi-periodic system, as time passes, two close initial conditions are *linearly* diverging, while in a chaotic system the divergence is exponential.

**Theorem 3.** *Assume that an RNN $F_{\boldsymbol{\theta}} \in \mathcal{R}$ (parameterized by $\boldsymbol{\theta}$) has a quasi-periodic attractor $\Gamma$ with $\mathcal{B}_\Gamma$ as its basin of attraction. Then, for every $\boldsymbol{z}_1 \in \mathcal{B}_\Gamma$*

$$\forall \, 0 < \epsilon < 1 \; \exists T_0 > 1 \; s.t. \; \forall T \geq T_0 \implies$$

$$(1 - \epsilon)^{T-1} < \left\| \frac{\partial \boldsymbol{z}_T}{\partial \boldsymbol{z}_1} \right\| < (1 + \epsilon)^{T-1}. \tag{14}$$

*Proof.* See Appx. A.2.3. $\qquad \square$

According to Theorem 3, for every orbit converging to a quasi-periodic attractor, the Jacobians $\frac{\partial \boldsymbol{z}_T}{\partial \boldsymbol{z}_t}$ may diverge or vanish as $T \to \infty$, but this will *not* occur exponentially fast as $T \to \infty$. Thus, even for bounded non-chaotic RNNs we may sometimes stumble into the problem of diverging gradients. Although this may be a less common scenario, we point out it may occur if we train RNNs on real data from oscillatory systems with incommensurate frequencies, as for instance encountered in electronic engineering.

In Appx. A.3.2 we have collected further mathematical results on the connection between RNN dynamics and loss gradients that hold regardless of the RNN's limiting behavior.

## 4   Empirical evaluation

Our theoretical results imply that chaotic time series pose a principle challenge for RNN training that cannot easily be circumvented through specifically designed architectures, constraints, or regularization criteria. If the underlying DS we aim to capture is chaotic, loss gradients propagated back in time will inevitably explode. Hence we need to curtail gradients in an ideal way. The issue arises especially in scientific ML where time series from chaotic systems are ubiquitous and the aim is to *reconstruct* the generating DS with its limiting behavior. Thus, our exposition will focus on this area.

### 4.1   Training on systems with exploding gradients by sparse teacher forcing

To illustrate the connections between theory and RNN training, we revive the old idea of TF [95] as a mechanism for truncating error gradients and keeping model-generated trajectories on track while training. However, we would like to do this such that important information about the system dynamics does not get lost, for which Lyapunov theory offers some guidance. Specifically, we should not force the system back onto the true trajectory all or most of the time (as in "classical TF"), but should effectively "re-calibrate" it only at certain time points chosen wisely according to the system's local divergence rates. This procedure will be referred to as *sparsely forced BPTT* in the following. Assume we want to train an RNN with hidden states $\boldsymbol{z}_t \in \mathbb{R}^M$ and linear (or affine) output layer on a time-series $\{\boldsymbol{x}_1, \boldsymbol{x}_2, \cdots, \boldsymbol{x}_T\}$ generated by a chaotic system.[3] The linear output layer $\hat{\boldsymbol{x}}_t = \boldsymbol{B}\boldsymbol{z}_t$, $\boldsymbol{B} \in \mathbb{R}^{N \times M}$, maps the RNN hidden states into the observation space. This allows us to modify the original TF procedure by constructing a control series $\{\tilde{\boldsymbol{z}}_1, \tilde{\boldsymbol{z}}_2, \cdots, \tilde{\boldsymbol{z}}_T\}$ from the observations by "inverting" the linear output mapping[4]

$$\tilde{\boldsymbol{z}}_t = (\boldsymbol{B}^{\mathsf{T}}\boldsymbol{B})^{-1}\boldsymbol{B}^{\mathsf{T}}\boldsymbol{x}_t. \tag{15}$$

The idea is to supply this control signal only sparsely, separated by the learning interval $\tau$ between consecutive forcings. Hence, defining $\mathcal{T} = \{n\tau + 1\}_{n \in \mathbb{N}_0}$ as the set of all time points at which we force the RNN onto the 'true' values, the RNN updates can be written as

$$\boldsymbol{z}_{t+1} = \begin{cases} RNN(\tilde{\boldsymbol{z}}_t) & \text{if } t \in \mathcal{T} \\ RNN(\boldsymbol{z}_t) & \text{else} \end{cases}. \tag{16}$$

This forcing is applied *after* calculation of the loss, such that $\mathcal{L}_t = \|\boldsymbol{x}_t - \boldsymbol{B}\boldsymbol{z}_t\|_2^2$ irrespective of whether $t$ is in $\mathcal{T}$ or not (and of course it is applied only during training, not at test time!). Replacing hidden states $\boldsymbol{z}_t$ with their teacher-forced signals $\tilde{\boldsymbol{z}}_t$ simply breaks divergence between true and predicted trajectories at time points $t \in \mathcal{T}$, and also cuts off the Jacobians by breaking the temporal contingency (for details see Appx. A.7). The learning interval $\tau$ hence controls how many time steps are included in the gradient calculation and has to be chosen with care such as to balance the effects of exploding gradients vs. those of losing relevant time scales and long-term dependencies. While it is general wisdom that an optimal batch size will facilitate training,[5] the point here is thus much

---

[3]Note that in DS reconstruction one usually considers the data as observations (*unsupervised* problem).

[4]To ensure invertibility, one could add a regularizer $\lambda \mathbf{I}$ to $\boldsymbol{B}^{\mathsf{T}}\boldsymbol{B}$ in eqn. (15), as in ridge regression, but we did not find this necessary in any of our examples.

[5]It is also reminiscent of truncated BPTT, but with the all-important differences that we suggest 1) a theoretically informed choice of the optimal 'truncation length' (forcing interval) and 2) a specific procedure for replacing current latent states by control values. As shown in sect. 4.2 & 4.3, both these aspects are indeed crucial to avoid diverging gradients and trajectories whilst not loosing relevant longer time scales.

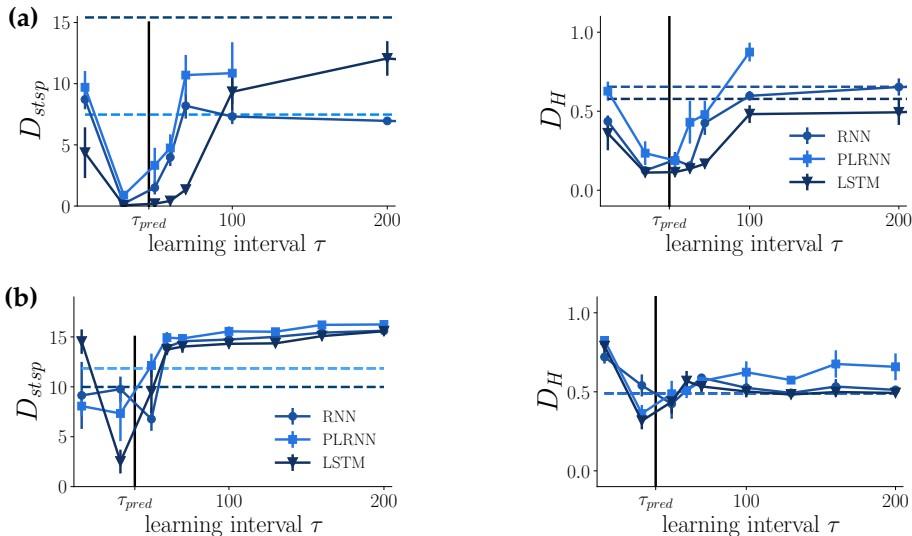

Figure 2: Overlap in attractor geometry ($D_{stsp}$, lower = better) and dimension-wise comparison of power-spectra ($D_H$, lower = better) against learning interval $\tau$ for (a) the Lorenz and (b) the chaotically forced Duffing oscillator. Continuous lines = sparsely forced BPTT. Dashed lines = classical BPTT with gradient clipping. Prediction time indicated vertically in black.

more specific: Ideally $\tau$ should be chosen in accordance with the system's Lyapunov spectrum, for instance based on the *predictability time* [7]

$$\tau_{\text{pred}} = \frac{\ln 2}{\lambda_{\max}}. \tag{17}$$

Various open-source packages exist for calculating the maximal Lyapunov exponent $\lambda_{\max}$ from empirical time series data (e.g., Julia: DynamicalSystems.jl [12], C++: TISEAN [30]), potentially after delay-embedding the data (see Appx. A.4 for more details). Note that this needs to be done on the *empirical* data and only once *before* RNN training, as $\lambda_{\max}$ is an invariant characteristic of the empirical system that we are aiming to reconstruct by our RNN (i.e., after successful training the RNN should have the same invariant properties as the underlying DS). We also emphasize that such a simple recipe for addressing the exploding gradient problem is based on modifying the training routine, and is thus in principle applicable to any model architecture.

### 4.2 Example 1: Lorenz system and externally forced Duffing oscillator in chaotic regime

Let us illustrate these ideas on two classical textbook examples of chaotic DS, the chaotic Lorenz attractor as an autonomous system, and the chaotically forced Duffing oscillator as an example with explicit external input (see Appx. A.4 for details). Trajectories were repeatedly drawn from these systems, on which we trained a PLRNN, a vanilla RNN with tanh activation function, and a LSTM by stochastic gradient descent (SGD) to minimize the MSE loss between predicted and actual observations. As optimizer we used Adam [47] from PyTorch [68] with a learning rate of $0.001$. For all models, training proceeded solely by *sparsely forced BPTT* and did not employ gradient clipping or any other technique that may interfere with optimal loss truncation.

In nonlinear DS reconstruction, we are mainly interested in reproducing *invariant* properties of the underlying system such as the attractor geometry (or topology [78, 82]) or the frequency composition (i.e., time-independent properties), while measures like ahead-prediction errors are less meaningful especially on chaotic time series [50, 96]. Thus, in evaluating training performance, here we follow Koppe et al. [50] in using a Kullback-Leibler divergence $D_{stsp}$ to quantify the agreement between observed and generated probability distributions across state-space to asses the overlap in attractor geometry (Appx. A.5). Moreover, we calculate the dimension-wise Hellinger distance $D_H$ between power spectra to quantify the temporal agreement of the observed and generated time-series (Appx. A.5).

Fig. 2 shows the dependence of the reconstruction quality on the learning interval $\tau$ for all RNN architectures on (a) the Lorenz and (b) the externally forced Duffing system. Fig. 3 provides particular examples of reconstructions for $\tau$ chosen too small, too large, or about right.

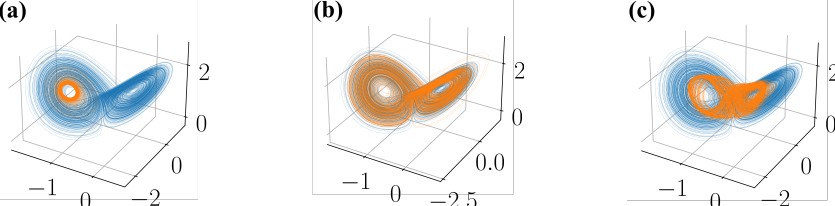

Figure 3: Lorenz attractor (blue) and example reconstructions by an LSTM (orange) trained with a learning interval (a) chosen too small ($\tau = 5$), (b) chosen optimally ($\tau = 30$), and (c) chosen too large ($\tau = 200$). See Fig. 14 for a vanilla RNN example.

For all models we find a system-dependent range for the optimal learning interval that agrees well with the predictability time defined in eqn. (17), where estimates for the maximal Lyapunov exponent were taken from the literature [24, 72]. As a reference, dashed lines represent the reconstruction performance for all architectures when trained with classical BPTT and gradient clipping. The training procedure was the same as for sparsely forced BPTT, except that instead of supplying a control-signal, gradients were normalized to 1 prior to each parameter update (see Fig. 13 for a more systematic evaluation of different clipping procedures and thresholds). As evidenced by the much worse performance, gradient clipping does not effectively address the EVGP, *even for LSTMs*. As further shown in Fig. 10 in Appx. A.6.4, using the optimal (or any other) window length $\tau$ but resetting the initial condition for each chunk to either zero or the last forward-iterated state $F_{\boldsymbol{\theta}}(\boldsymbol{z}_{t-1})$ (instead of its control value $\tilde{\boldsymbol{z}}_t$) equally destroys performance. This suggests that neither mere gradient normalization nor simple windowing are sufficient, but will wipe out essential information about the dynamics.

In Appx. A.6 we collect further results on the chaotic Rössler attractor (Fig. 6 & 7), high-dimensional Mackey-Glass equations (Fig. 8), and the Lorenz attractor with partial observations (Fig. 9).

## 4.3 Example 2: Chaotic weather data

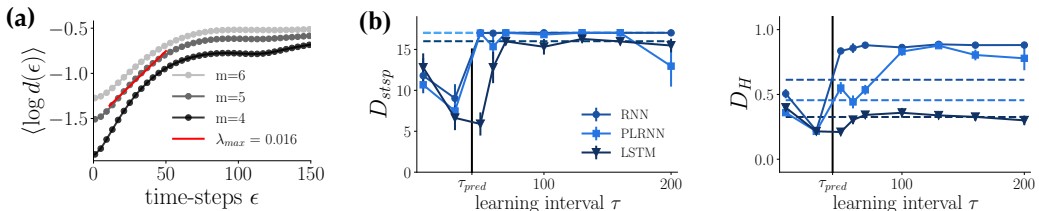

Figure 4: (a) The maximal Lyapunov exponent was determined as the slope of the average log-divergence of nearest neighbors in embedding space ($m$ = embedding dimension). (b) Reconstruction quality assessed by attractor overlap (lower = better) and dimension-wise comparison of power-spectra ($D_H$, lower = better). Black vertical lines = $\tau_{\text{pred}}$.

As for an empirical example, we trained all RNNs (vanilla RNN, PLRNN, LSTM) on a temperature time series recorded at the Weather Station at the Max Planck Institute for Biogeochemistry in Jena, Germany. To expose the chaotic behavior and obtain a robust estimate of the maximal Lyapunov exponent, trends and yearly cycles were removed, and nonlinear noise-reduction was performed ([43]; Appx. A.4). The maximal Lyapunov exponent was determined with the *TISEAN* package [30], as shown in Figure 4 (a). The value obtained is in close agreement with the literature ($\lambda_{max} \approx 0.017$ [62]).

Figure 4 shows that also for these empirical data the optimal training interval $\tau$ agrees well with the predictability time, eqn. (17), for all trained RNNs. Furthermore, as was the case for the DS benchmarks, gradient clipping was not able to satisfactorily tackle the EVGP, even when paired with

architectures like LSTMs explicitly designed for alleviating this problem. Similar results are reported for another real-world dataset, electroencephalogram (EEG) recordings, in Appx. A.6.5.

## 5    Discussion and conclusions

In this paper we proved that RNN dynamics and loss gradients are intimately related for all major types of RNNs and activation functions. If the RNN is "well behaved" in the sense that its dynamics converges to a fixed point or cycle, loss gradients will remain bounded, and established remedies [35, 80] can be used to refrain them from vanishing. However, if the dynamics are chaotic, gradients will always explode. This constitutes a *principle* problem in RNN training that cannot easily be mastered through architectural design or gradient clipping. This is because to avoid exploding gradients while training on time series from chaotic systems, one either needs to constrain the RNN so much that chaotic behavior is completely disabled to begin with (i.e., ultimately by forcing all Lyapunov exponents to be smaller or equal to zero), implying a very poor fit to such data. Or one needs to be a bit more lenient and thereby allow for the possibility of exploding loss gradients (as LSTMs or PLRNNs in fact do). This problem is furthermore practically highly relevant, as most time series we encounter in nature, and many from man-made systems as well, are inherently chaotic.

While we do not offer a full solution to this problem here, we suggest it might be tackled in training by taking a system's local divergence rates as measured through the Lyapunov spectrum into account. Hence, rather than conquering the EVGP by structural design or specific constraints or regularization terms, we recommend to put the focus more on the training process itself. We illustrated this point empirically using *sparsely forced BPTT*, a training technique that pulls trajectories back on track at times determined by the maximal Lyapunov exponent. Doing so leads to optimal reconstruction results for a variety of simulated and real-world benchmarks, regardless of the specific RNN architecture employed in training.

As noted in sect. 4.1, fairly standard packages are available for computing maximal Lyapunov exponents from data. Some background knowledge, as provided in classical textbooks (e.g. Ch. 5 in [42]), may be required for properly reading the output from these packages: Essentially, one would be looking be for a linear scaling region as in Figs. 4a & 12a, ignoring both the initial noise transient as well as the plateau caused by reaching the full attractor extent. If unsure about the exact value, a moderate amount of jittering around the estimated mean value may help (see Appx. Fig. 15). A further interesting direction for improvement might be to regulate the forcing interval through an annealing procedure [1, 2], for instance starting at $\tau = 1$ and ramping up to $\tau = \tau_{pred}$ throughout training, similar as in adaptive schemes [5].[6] The idea here would be to first get the short-term behavior right, and then challenge the system more and more for longer time spans until the predictability time is reached.

We stress that our goal here above all was to provide a mathematically grounded perspective on the problem, with the empirical section focused on elucidating the practical implications of the theoretical results. We believe that a more thorough theoretical understanding is important and needed for guiding future research into more powerful training procedures that avoid exploding gradients *without* compromising expressiveness. In our application examples, we developed the case from the perspective of scientific machine learning, which by now is a broad area in its own right with huge societal relevance (e.g., climate or epidemiological time series), and where the reconstruction of geometrical or topological (invariant) properties is important, beyond mere prediction. Nevertheless, we believe that our theoretical results will also have implications for other domains, like NLP [36]. While scientific time series problems traditionally have been extensively considered from a DS perspective (e.g., [42]), much more groundwork is needed, however, in areas like NLP, where, for instance, it may not even be immediately clear how to best define a Lyapunov spectrum.

All code from this paper is available at `https://github.com/DurstewitzLab/ChaosRNN`.

## Acknowledgements

This work was funded by the German Research Foundation (DFG) under Germany's Excellence Strategy – EXC-2181 – 390900948 (STRUCTURES), and through grant Du 354/10-1 to DD.

---

[6]We thank one of the referees for pointing this out.

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
