# A  Appendix

## A.1  Theorems: Preliminaries

### A.1.1  Transforming non-autonomous into autonomous discrete-time DS

Following [97], and based on similar reasoning as for continuous time (ODE-based) DS [3, 71], let us consider the non-autonomous discrete-time DS

$$\boldsymbol{x}_{t+1} = F(\boldsymbol{x}_t, t), \quad \boldsymbol{x} \in \mathbb{R}^n. \tag{18}$$

Defining $\boldsymbol{z}_t = (\boldsymbol{x}_t, t)^\mathsf{T}$ and $G(\boldsymbol{z}_t) = (F(\boldsymbol{x}_t, t), t+1)^\mathsf{T}$, system (18) can be rewritten as the autonomous system

$$\boldsymbol{z}_{t+1} = G(\boldsymbol{z}_t), \quad \boldsymbol{z} \in \mathbb{R}^{n+1}. \tag{19}$$

Hence, in all our theoretical treatment we can confine our attention to systems of the form (19).

### A.1.2  RNN derivatives

Considering the loss function $\mathcal{L} = \sum_{t=1}^{T} \mathcal{L}_t$ of an RNN $F_{\boldsymbol{\theta}} \in \mathcal{R}$ parameterized by $\boldsymbol{\theta}$, we have

$$\frac{\partial \mathcal{L}}{\partial \theta} = \sum_{t=1}^{T} \frac{\partial \mathcal{L}_t}{\partial \theta}, \tag{20}$$

where

$$\frac{\partial \mathcal{L}_t}{\partial \theta} = \frac{\partial \mathcal{L}_t}{\partial \boldsymbol{z}_t} \frac{\partial \boldsymbol{z}_t}{\partial \theta}. \tag{21}$$

The tangent vector $\frac{\partial \boldsymbol{z}_T}{\partial \theta}$ has the form

$$\frac{\partial \boldsymbol{z}_T}{\partial \theta} = \frac{\partial^+ \boldsymbol{z}_T}{\partial \theta} + \sum_{t=1}^{T-2} \left( \prod_{r=0}^{t-1} \boldsymbol{J}_{T-r} \right) \frac{\partial^+ \boldsymbol{z}_{T-t}}{\partial \theta}, \tag{22}$$

where $\partial^+$ denotes the immediate partial derivative. Since for an RNN $F_{\boldsymbol{\theta}} \in \mathcal{R}$ the activation function is element-wise, with $\theta$ the $m$-th element of a parameter vector $\boldsymbol{\theta}$ (or belonging to the $m$-th row of a parameter matrix $\boldsymbol{\theta}$), we have

$$\frac{\partial^+ \boldsymbol{z}_T}{\partial \theta} = \begin{pmatrix} 0 & \cdots & 0 & \frac{\partial^+ z_{m,T}}{\partial \theta} & 0 & \cdots & 0 \end{pmatrix}^\mathsf{T}. \tag{23}$$

For instance, let $\boldsymbol{\theta} = \boldsymbol{W}$ be a weight matrix, then

$$\frac{\partial \mathcal{L}}{\partial \boldsymbol{W}} = \begin{pmatrix} \frac{\partial \mathcal{L}}{\partial w_{11}} & \frac{\partial \mathcal{L}}{\partial w_{12}} & \cdots & \frac{\partial \mathcal{L}}{\partial w_{1M}} \\ \frac{\partial \mathcal{L}}{\partial w_{21}} & \frac{\partial \mathcal{L}}{\partial w_{22}} & \cdots & \frac{\partial \mathcal{L}}{\partial w_{2M}} \\ \vdots & & & \\ \frac{\partial \mathcal{L}}{\partial w_{M1}} & \frac{\partial \mathcal{L}}{\partial w_{M2}} & \cdots & \frac{\partial \mathcal{L}}{\partial w_{MM}} \end{pmatrix}. \tag{24}$$

In this case, for the standard RNN we have

$$\frac{\partial^+ \boldsymbol{z}_T}{\partial w_{mk}} = \begin{pmatrix} 0 & \cdots & 0 & z_{k,T-1}\, \xi_{mk}(\boldsymbol{z}_{T-1}) & 0 & \cdots & 0 \end{pmatrix}^\mathsf{T} = \mathbf{1}_{(m,k)}\, \xi_{mk}(\boldsymbol{z}_{T-1})\, z_{T-1}, \tag{25}$$

where $\xi_{mk}(\boldsymbol{z}_{T-1}) = f'_{w_{m,k}} \left( \sum_{j=1}^{M} w_{mj}\, z_{j,T-1} + \sum_{j=1}^{M} b_{mj}\, s_{j,T} + h_m \right)$, and $f'_{w_{m,k}}$ stands for the derivative of $f$ with respect to $w_{m,k}$.

Therefore, for standard RNNs, (22) becomes

$$\frac{\partial \boldsymbol{z}_T}{\partial w_{mk}} = \mathbf{1}_{(m,k)}\, \xi_{mk}(\boldsymbol{z}_{T-1})\, z_{T-1} + \sum_{t=1}^{T-2} \left( \prod_{r=0}^{t-1} \boldsymbol{J}_{T-r} \right) \mathbf{1}_{(m,k)}\, \xi_{mk}(\boldsymbol{z}_{T-t-1})\, z_{T-t-1}. \tag{26}$$

### A.1.3 Piecewise-linear RNN (PLRNN)

The PLRNN has the generic form [50, 80]

$$z_t = F(z_{t-1}) = A z_{t-1} + W \phi(z_{t-1}) + C s_t + h + \varepsilon_t, \tag{27}$$

where $\phi(z_{t-1}) = \max(z_{t-1}, 0)$ is the element-wise rectified linear unit (ReLU) function, $z_t \in \mathbb{R}^M$ is the neural state vector, $A \in \mathbb{R}^{M \times M}$ is a diagonal matrix of auto-regression weights, $W \in \mathbb{R}^{M \times M}$ is a matrix of connection weights, $h \in \mathbb{R}^M$ is the bias vector, $s_t \in \mathbb{R}^K$ the external input weighted by $C \in \mathbb{R}^{M \times K}$, and $\varepsilon_t \sim \mathcal{N}(0, \Sigma)$ a Gaussian noise term with diagonal covariance matrix $\Sigma$.

Equation (27) can be rewritten as

$$z_t = (A + W D_{\Omega(t-1)}) z_{t-1} + C s_t + h + \varepsilon_t =: W_{\Omega(t-1)} z_{t-1} + C s_t + h + \varepsilon_t, \tag{28}$$

where $D_{\Omega(t)} := \operatorname{diag}(d_{\Omega(t)})$ is a diagonal matrix with $d_{\Omega(t)} := (d_1, d_2, \cdots, d_M)$ an indicator vector such that $d_m(z_{m,t}) =: d_m = 1$ whenever $z_{m,t} > 0$, and zeros otherwise.

For the PLRNN (28) we have

$$J_t = \frac{\partial z_t}{\partial z_{t-1}} = W_{\Omega(t-1)}, \tag{29}$$

and $\left\| W_{\Omega(t-1)} \right\| \leq \|A\| + \|W\|$.

Furthermore, the derivatives (22) for the PLRNN (28) are

$$\frac{\partial z_T}{\partial w_{mk}} = \mathbf{1}_{(m,k)} D_{\Omega(T-1)} z_{T-1} + \sum_{j=2}^{T-1} \left( \prod_{i=1}^{j-1} W_{\Omega(T-i)} \right) \mathbf{1}_{(m,k)} D_{\Omega(T-j)} z_{T-j}. \tag{30}$$

### A.1.4 Long Short-Term Memory (LSTM)

The LSTM is defined by the equations

$$i_t = \sigma \big( W_{ii} s_t + W_{hi} h_{t-1} + b_i \big)$$

$$f_t = \sigma \big( W_{if} s_t + W_{hf} h_{t-1} + b_f \big)$$

$$g_t = \tanh \big( W_{ig} s_t + W_{hg} h_{t-1} + b_g \big)$$

$$o_t = \sigma \big( W_{io} s_t + W_{ho} h_{t-1} + b_o \big)$$

$$c_t = f_t \odot c_{t-1} + i_t \odot g_t$$

$$h_t = o_t \odot \tanh (c_t) \tag{31}$$

where $\{s_t\}$ is the input sequence, $W$ denotes weight matrices, $b$ bias terms, $i_t, f_t, g_t, o_t$ demonstrate the input, forget, cell, and output gates, $h_t$ and $c_t$ are the hidden and cell states at time $t$ respectively, $\sigma$ is the sigmoid activation function, and $\odot$ represents the element-wise (Hadamard) product (see [29, 35, 90] for further information on LSTMs).

Defining $z_t := (h_t, c_t)^\mathsf{T}$, the LSTM (31) can be represented as the first-order recursive map

$$z_t = F_{\boldsymbol{\theta}}(z_{t-1}) = \begin{pmatrix} o_t \odot \tanh (f_t \odot c_{t-1} + i_t \odot g_t) \\ f_t \odot c_{t-1} + i_t \odot g_t \end{pmatrix}. \tag{32}$$

The term $\frac{\partial \mathcal{L}_t}{\partial \theta}$ in (20) for some LSTM parameter $\theta$ can be written as

$$\frac{\partial \mathcal{L}_t}{\partial \theta} = \sum_{r=1}^{t} \frac{\partial \mathcal{L}_t}{\partial h_t} \frac{\partial h_t}{\partial z_t} \frac{\partial z_t}{\partial z_r} \frac{\partial z_r}{\partial \theta}. \tag{33}$$

A necessary condition for LSTMs to have a chaotic orbit is given by:

**Proposition 1.** *Let the LSTM given by (37) have a chaotic attractor $\Gamma^*$ with $\mathcal{B}_{\Gamma^*}$ as its basin of attraction. Then for every $\boldsymbol{z}_1 = (\boldsymbol{h}_1, \boldsymbol{c}_1)^\top \in \mathcal{B}_{\Gamma^*}$*

$$\gamma := \lim_{T\to\infty} \sqrt[T]{\left\| \begin{pmatrix} \frac{\partial \boldsymbol{h}_T}{\partial \boldsymbol{h}_1} & \frac{\partial \boldsymbol{h}_T}{\partial \boldsymbol{c}_1} \\ \frac{\partial \boldsymbol{c}_T}{\partial \boldsymbol{h}_1} & \frac{\partial \boldsymbol{c}_T}{\partial \boldsymbol{c}_1} \end{pmatrix} \right\|} > 1. \tag{34}$$

*Proof.* The Jacobian matrix of (32) for $t > 1$ can be written in the block form

$$\frac{\partial \boldsymbol{z}_t}{\partial \boldsymbol{z}_{t-1}} = J_t = \begin{pmatrix} \frac{\partial \boldsymbol{h}_t}{\partial \boldsymbol{h}_{t-1}} & \frac{\partial \boldsymbol{h}_t}{\partial \boldsymbol{c}_{t-1}} \\ \frac{\partial \boldsymbol{c}_t}{\partial \boldsymbol{h}_{t-1}} & \frac{\partial \boldsymbol{c}_t}{\partial \boldsymbol{c}_{t-1}} \end{pmatrix}. \tag{35}$$

Further, due to the chain rule, we have

$$J_t J_{t-1} = \begin{pmatrix} \frac{\partial \boldsymbol{h}_t}{\partial \boldsymbol{h}_{t-1}}\frac{\partial \boldsymbol{h}_{t-1}}{\partial \boldsymbol{h}_{t-2}} + \frac{\partial \boldsymbol{h}_t}{\partial \boldsymbol{c}_{t-1}}\frac{\partial \boldsymbol{c}_{t-1}}{\partial \boldsymbol{h}_{t-2}} & \frac{\partial \boldsymbol{h}_t}{\partial \boldsymbol{h}_{t-1}}\frac{\partial \boldsymbol{h}_{t-1}}{\partial \boldsymbol{c}_{t-2}} + \frac{\partial \boldsymbol{h}_t}{\partial \boldsymbol{c}_{t-1}}\frac{\partial \boldsymbol{c}_{t-1}}{\partial \boldsymbol{c}_{t-2}} \\ \frac{\partial \boldsymbol{c}_t}{\partial \boldsymbol{h}_{t-1}}\frac{\partial \boldsymbol{h}_{t-1}}{\partial \boldsymbol{h}_{t-2}} + \frac{\partial \boldsymbol{c}_t}{\partial \boldsymbol{c}_{t-1}}\frac{\partial \boldsymbol{c}_{t-1}}{\partial \boldsymbol{h}_{t-2}} & \frac{\partial \boldsymbol{c}_t}{\partial \boldsymbol{h}_{t-1}}\frac{\partial \boldsymbol{h}_{t-1}}{\partial \boldsymbol{c}_{t-2}} + \frac{\partial \boldsymbol{c}_t}{\partial \boldsymbol{c}_{t-1}}\frac{\partial \boldsymbol{c}_{t-1}}{\partial \boldsymbol{c}_{t-2}} \end{pmatrix}$$

$$= \begin{pmatrix} \frac{\partial \boldsymbol{h}_t}{\partial \boldsymbol{h}_{t-2}} & \frac{\partial \boldsymbol{h}_t}{\partial \boldsymbol{c}_{t-2}} \\ \frac{\partial \boldsymbol{c}_t}{\partial \boldsymbol{h}_{t-2}} & \frac{\partial \boldsymbol{c}_t}{\partial \boldsymbol{c}_{t-2}} \end{pmatrix}, \tag{36}$$

and by induction we obtain

$$\frac{\partial \boldsymbol{z}_t}{\partial \boldsymbol{z}_1} = J_t J_{t-1} J_{t-2} \cdots J_2 = \begin{pmatrix} \frac{\partial \boldsymbol{h}_t}{\partial \boldsymbol{h}_1} & \frac{\partial \boldsymbol{h}_t}{\partial \boldsymbol{c}_1} \\ \frac{\partial \boldsymbol{c}_t}{\partial \boldsymbol{h}_1} & \frac{\partial \boldsymbol{c}_t}{\partial \boldsymbol{c}_1} \end{pmatrix}. \tag{37}$$

Now assume that (32) has a chaotic orbit given by

$$\Gamma^* = \{\boldsymbol{z}_1^*, \boldsymbol{z}_2^*, \cdots, \boldsymbol{z}_T^*, \cdots\}. \tag{38}$$

According to (37), the largest Lyapunov exponent of $\Gamma^*$ is given by

$$\lambda_{\Gamma^*} = \lim_{T\to\infty} \frac{1}{T} \ln \left\| J_T^* J_{T-1}^* \cdots J_2^* \right\| = \lim_{T\to\infty} \frac{1}{T} \ln \left\| \begin{pmatrix} \frac{\partial \boldsymbol{h}_T^*}{\partial \boldsymbol{h}_1^*} & \frac{\partial \boldsymbol{h}_T^*}{\partial \boldsymbol{c}_1^*} \\ \frac{\partial \boldsymbol{c}_T^*}{\partial \boldsymbol{h}_1^*} & \frac{\partial \boldsymbol{c}_T^*}{\partial \boldsymbol{c}_1^*,} \end{pmatrix} \right\|.$$

Since $\Gamma^*$ is chaotic, so $\lambda_{\Gamma^*} > 0$, which gives

$$\lim_{T\to\infty} \sqrt[T]{\left\| \begin{pmatrix} \frac{\partial \boldsymbol{h}_T^*}{\partial \boldsymbol{h}_1^*} & \frac{\partial \boldsymbol{h}_T^*}{\partial \boldsymbol{c}_1^*} \\ \frac{\partial \boldsymbol{c}_T^*}{\partial \boldsymbol{h}_1^*} & \frac{\partial \boldsymbol{c}_T^*}{\partial \boldsymbol{c}_1^*} \end{pmatrix} \right\|} > 1. \tag{39}$$

Based on Oseledec's multiplicative ergodic Theorem, (39) holds for every $\boldsymbol{z}_1 \in \mathcal{B}_{\Gamma^*}$. This completes the proof. $\square$

### A.1.5 Gated Recurrent Unit (GRU)

A GRU network is defined by the equations

$$\boldsymbol{z}_t = \sigma\big(\boldsymbol{W}_z\,\boldsymbol{s}_t + \boldsymbol{U}_z\boldsymbol{h}_{t-1} + \boldsymbol{b}_z\big)$$

$$\boldsymbol{r}_t = \sigma\big(\boldsymbol{W}_r\,\boldsymbol{s}_t + \boldsymbol{U}_r\boldsymbol{h}_{t-1} + \boldsymbol{b}_r\big)$$

$$\boldsymbol{h}_t = (1 - \boldsymbol{z}_t) \odot \tanh\big(\boldsymbol{W}_h\,\boldsymbol{s}_t + \boldsymbol{U}_h(\boldsymbol{r}_t \odot \boldsymbol{h}_{t-1}) + \boldsymbol{b}_h\big) + \boldsymbol{z}_t \odot \boldsymbol{h}_{t-1}, \tag{40}$$

where $\boldsymbol{r}_t$ represents the reset gate, $\boldsymbol{z}_t$ the update gate, $\boldsymbol{s}_t$ and $\boldsymbol{h}_t$ denote the inputs and the hidden state respectively, $\boldsymbol{W}_z, \boldsymbol{W}_r, \boldsymbol{W}_h \in \mathbb{R}^{M \times N}$ and $\boldsymbol{U}_z, \boldsymbol{U}_r, \boldsymbol{U}_h \in \mathbb{R}^{M \times M}$ are weight matrices, $\boldsymbol{b}_z, \boldsymbol{b}_r, \boldsymbol{b}_h \in \mathbb{R}^M$ are bias vectors, and $\sigma$ is the element-wise logistic sigmoid function (for more details about GRUs see [10]).

### A.1.6 Unitary evolution RNN (uRNN)

The uRNN, proposed in [4], is defined as the nonlinear DS

$$\boldsymbol{z}_t = \sigma_{\boldsymbol{b}}\big(\boldsymbol{W}\boldsymbol{z}_{t-1} + \boldsymbol{V}\boldsymbol{s}_t\big), \tag{41}$$

for which $\boldsymbol{W} \in U(M)$ is an unitary matrix, $\boldsymbol{V} \in \mathbb{C}^{M \times N}$, $\boldsymbol{b} \in \mathbb{R}^M$ is the bias parameter, $\boldsymbol{s}_t$ is the real- or complex-valued input of dimension $N$, and

$$[\sigma_{\boldsymbol{b}}(\boldsymbol{z})]_i = [\sigma_{\text{modReLU}}(\boldsymbol{z})]_i = \begin{cases} \big(|z_i| + b_i\big)\frac{z_i}{|z_i|} & \text{if } |z_i| + b_i \geq 0 \\ 0 & \text{if } |z_i| + b_i < 0 \end{cases}. \tag{42}$$

**Proposition 2.** *The uRNN given by (41) cannot have any chaotic orbit.*

*Proof.* For any arbitrary orbit $\mathcal{O}_{\boldsymbol{z}_1}$ of (41) we have

$$\|J_T J_{T-1} \cdots J_2\| = \left\|\prod_{k=0}^{T-2} \boldsymbol{D}_{T-k} \boldsymbol{W}^{\mathsf{T}}\right\|, \tag{43}$$

where $\boldsymbol{D}_t = diag\big(\sigma_{\boldsymbol{b}}'\big(\boldsymbol{W}\boldsymbol{z}_{t-1} + \boldsymbol{V}\boldsymbol{s}_t\big)\big)$. Since $\boldsymbol{W}$ is unitary and so a norm preserving matrix, it is concluded that

$$\left\|\prod_{k=0}^{T-2} \boldsymbol{D}_{T-k} \boldsymbol{W}^{\mathsf{T}}\right\| \leq \prod_{k=0}^{T-2} \|\boldsymbol{D}_{T-k} \boldsymbol{W}^{\mathsf{T}}\| = \prod_{k=0}^{T-2} \|\boldsymbol{D}_{T-k}\| = 1, \tag{44}$$

which implies

$$\lambda_{max} = \lim_{T \to \infty} \frac{1}{T} \ln \|J_T J_{T-1} \cdots J_2\| \leq 0. \tag{45}$$

This rules out the existence of chaos (since $\lambda_{max} > 0$ is a necessary condition for $\mathcal{O}_{\boldsymbol{z}_1}$ to be chaotic). □

Note that, more generally, any RNN which is constrained such as to exhibit global convergence to a fixed point or cycle, by definition must have a maximum Lyapunov exponent $\lambda_{max} \leq 0$ (in accordance with Theorem 1), hence cannot exhibit chaotic behavior by definition.

### A.2 Theorems: Proofs

### A.2.1 Proof of theorem 1, parts (ii) & (iii)

*Proof.* $(ii)$ If $\boldsymbol{J}$ is the Jordan normal form of $\prod_{s=0}^{k-1} J_{t^{*k}-s}$, then $\prod_{s=0}^{k-1} J_{t^{*k}-s} = \boldsymbol{P}\boldsymbol{J}\boldsymbol{P}^{-1}$, where

$$\boldsymbol{J} = \begin{pmatrix} \boldsymbol{J}_{m_1}(\lambda_1) & 0 & 0 & \cdots & 0 \\ 0 & \boldsymbol{J}_{m_2}(\lambda_2) & 0 & \cdots & 0 \\ \vdots & \cdots & \ddots & \cdots & \vdots \\ 0 & \cdots & 0 & \boldsymbol{J}_{m_{p-1}}(\lambda_{p-1}) & 0 \\ 0 & \cdots & \cdots & 0 & \boldsymbol{J}_{m_p}(\lambda_p) \end{pmatrix}, \tag{46}$$

and $m_i$ is the algebraic multiplicity of each eigenvalue $\lambda_i$. Since $\rho\big(\prod_{s=0}^{k-1} J_{t^{*k}-s}\big) < 1$, so the eigenvalue $\lambda_i$ associated with each Jordan block satisfies $|\lambda_i| < 1$ $(i = 1, \cdots, p)$. Moreover, every $m_i \times m_i$ Jordan block has the form

$$\boldsymbol{J}_{m_i}(\lambda_i) = \begin{pmatrix} \lambda_i & 1 & 0 & \cdots & 0 \\ 0 & \lambda_i & 1 & \cdots & 0 \\ \vdots & \vdots & \ddots & \ddots & \vdots \\ 0 & 0 & \cdots & \lambda_i & 1 \\ 0 & 0 & \cdots & 0 & \lambda_i \end{pmatrix}. \tag{47}$$

Accordingly

$$\left\| \left( \prod_{s=0}^{k-1} J_{t^{*k}-s} \right)^j \right\| = \left\| \boldsymbol{P} \boldsymbol{J}^j \boldsymbol{P}^{-1} \right\| \leq p \left\| \boldsymbol{J}^j \right\|, \tag{48}$$

in which $p = \|\boldsymbol{P}\| \|\boldsymbol{P}^{-1}\|$. Furthermore, for $j \in \mathbb{N}$, $\boldsymbol{J}^j$ is a block diagonal matrix of the form

$$\boldsymbol{J}^j = \begin{pmatrix} \boldsymbol{J}_{m_1}^j(\lambda_1) & 0 & 0 & \cdots & 0 \\ 0 & \boldsymbol{J}_{m_2}^j(\lambda_2) & 0 & \cdots & 0 \\ \vdots & \cdots & \ddots & \cdots & \vdots \\ 0 & \cdots & 0 & \boldsymbol{J}_{m_{p-1}}^j(\lambda_{p-1}) & 0 \\ 0 & \cdots & \cdots & 0 & \boldsymbol{J}_{m_p}^j(\lambda_p) \end{pmatrix}, \tag{49}$$

in which every $m_i \times m_i$ Jordan block has the form

$$\boldsymbol{J}_{m_i}^j(\lambda_i) = \begin{pmatrix} \lambda_i^j & \binom{j}{1} \lambda_i^{j-1} & \binom{j}{2} \lambda_i^{j-2} & \cdots & \binom{j}{m_i-1} \lambda_i^{j-m_i+1} \\ 0 & \lambda_i^j & \binom{j}{1} \lambda_i^{j-1} & \cdots & \binom{j}{m_i-2} \lambda_i^{j-m_i+2} \\ \vdots & \vdots & \ddots & \ddots & \vdots \\ 0 & 0 & \cdots & \lambda_i^j & \binom{j}{1} \lambda_i^{j-1} \\ 0 & 0 & \cdots & 0 & \lambda_i^j \end{pmatrix}. \tag{50}$$

In addition, for every block $\boldsymbol{J}_{m_i}^j(\lambda_i)$, we have

$$\left\| \boldsymbol{J}_{m_i}^j(\lambda_i) \right\| \leq \sqrt{m_i} \left\| \boldsymbol{J}_{m_i}^j(\lambda_i) \right\|_\infty = \sqrt{m_i} \sum_{q=1}^{m_i} \left| \left( \boldsymbol{J}_{m_i}^j(\lambda_i) \right)_{1q} \right|$$

$$= \sqrt{m_i} \sum_{q=1}^{m_i} \binom{j}{q-1} |\lambda_i|^{j-q+1} = |\lambda_i|^j \sqrt{m_i} \left( |\lambda_i|^{1-m_i} \sum_{q=1}^{m_i} \binom{j}{q-1} |\lambda_i|^{m_i-q} \right)$$

$$\leq |\lambda_i|^j \, j^{m_i} \sqrt{m_i} \left( |\lambda_i|^{1-m_i} \sum_{q=1}^{m_i} |\lambda_i|^{m_i-q} \right) =: |\lambda_i|^j \, j^{m_i} \, N_{\lambda_i}. \tag{51}$$

Moreover, for any $1 < \tilde{r}_i < \frac{1}{|\lambda_i|}$, there exists some $l_i$ such that $j^{m_i} < \tilde{r}_i^j$ for $j \geq l_i$. This means for $j \geq l_i$

$$\left\| \boldsymbol{J}_{m_i}^j(\lambda_i) \right\| \leq N_{\lambda_i} |\tilde{r}_i \lambda_i|^j, \tag{52}$$

such that $|\tilde{r}_i \lambda_i| = \tilde{r}_i |\lambda_i| < 1$.

Besides, for $\boldsymbol{J}^j = \boldsymbol{J}_{m_1}^j(\lambda_1) \oplus \boldsymbol{J}_{m_2}^j(\lambda_2) \oplus \cdots \oplus \boldsymbol{J}_{m_p}^j(\lambda_p)$

$$\left\| \boldsymbol{J}^j \right\| = \max_{1 \leq i \leq p} \left\| \boldsymbol{J}_{m_i}^j(\lambda_i) \right\| =: \left\| \boldsymbol{J}_m^j(\lambda) \right\|. \tag{53}$$

Hence, from (48), (52) and (53), it is deduced that for $j \geq l$

$$\left\| \left( \prod_{s=0}^{k-1} J_{t^{*k}-s} \right)^j \right\| \leq p \, N_\lambda \, |\tilde{r} \lambda|^j =: \bar{p} \, r^j, \tag{54}$$

in which $r = |\tilde{r} \lambda| < 1$.

Furthermore, let for $\Gamma_k$

$$\max_{T \geq 1}\Big\{ \|\boldsymbol{J}_T^*\| \Big\} = \max_{0 \leq s \leq k-1}\Big\{ \|J_{t^{*k}-s}\| \Big\} = \bar{m},$$

$$\max_{T \geq 1}\Big\{ \Big\|\frac{\partial^+ \boldsymbol{z}_T}{\partial \theta}\Big\| \Big\} = \max_{0 \leq s \leq k-1}\Big\{ \Big\|\frac{\partial^+ \boldsymbol{z}_{t^{*k}-s}}{\partial \theta}\Big\| \Big\} = \xi,$$

$$\max_{T \geq 1}\Big\{ \|\boldsymbol{z}_T\| \Big\} = \max_{0 \leq s \leq k-1}\Big\{ \|\boldsymbol{z}_{t^{*k}-s}\| \Big\} = \bar{q}. \tag{55}$$

Hence, defining $\boldsymbol{z}_0 = 0$, for this $k$-cycle

$$\Big\|\frac{\partial \boldsymbol{z}_T}{\partial \theta}\Big\| = \Big\|\frac{\partial^+ \boldsymbol{z}_T}{\partial \theta} + \sum_{t=1}^{T-2}\Big(\prod_{r=0}^{t-1} \boldsymbol{J}_{T-r}^*\Big)\frac{\partial^+ \boldsymbol{z}_{T-t}}{\partial \theta}\Big\|$$

$$= \Big\|\frac{\partial^+ \boldsymbol{z}_T}{\partial \theta} + \sum_{t=1}^{T-1}\Big(\prod_{r=0}^{t-1} \boldsymbol{J}_{T-r}^*\Big)\frac{\partial^+ \boldsymbol{z}_{T-t}}{\partial \theta}\Big\|$$

$$\leq \bar{q}\xi\Big(1 + \sum_{t=1}^{T-1}\Big\|\prod_{r=0}^{t-1} \boldsymbol{J}_{T-r}^*\Big\|\Big). \tag{56}$$

On the other hand, for $T = kj$, from (54) and (55) we have

$$\sum_{t=1}^{T-1}\Big\|\prod_{r=0}^{t-1} \boldsymbol{J}_{T-r}^*\Big\| = \sum_{t=1}^{kj-1}\Big\|\prod_{r=0}^{t-1} \boldsymbol{J}_{kj-r}^*\Big\| = \sum_{t=1}^{k-1}\Big\|\prod_{r=0}^{t-1} \boldsymbol{J}_{kj-r}^*\Big\| + \sum_{t=k}^{2k-1}\Big\|\prod_{r=0}^{t-1} \boldsymbol{J}_{kj-r}^*\Big\|$$

$$+ \sum_{t=2k}^{3k-1}\Big\|\prod_{r=0}^{t-1} \boldsymbol{J}_{kj-r}^*\Big\| + \cdots + \sum_{t=(j-2)k}^{(j-1)k-1}\Big\|\prod_{r=0}^{t-1} \boldsymbol{J}_{kj-r}^*\Big\| + \sum_{t=(j-1)k}^{kj-1}\Big\|\prod_{r=0}^{t-1} \boldsymbol{J}_{kj-r}^*\Big\|$$

$$= \sum_{t=1}^{k-1}\Big\|\prod_{r=0}^{t-1} \boldsymbol{J}_{kj-r}^*\Big\| + \sum_{i=2}^{j}\sum_{t=(i-1)k}^{ik-1}\Big\|\prod_{r=0}^{t-1} \boldsymbol{J}_{kj-r}^*\Big\|$$

$$\leq \big(\bar{m} + \bar{m}^2 + \cdots + \bar{m}^{k-1}\big) + \sum_{i=2}^{j}\bar{p}\big(1 + \bar{m} + \bar{m}^2 + \cdots + \bar{m}^{k-1}\big)r^{i-1}. \tag{57}$$

Thus, considering $\big(\bar{m} + \bar{m}^2 + \cdots + \bar{m}^{k-1}\big) = \mathcal{M}$, it is deduced that

$$\lim_{T \to \infty}\Big\|\frac{\partial \boldsymbol{z}_T}{\partial \theta}\Big\| = \lim_{j \to \infty}\Big\|\frac{\partial \boldsymbol{z}_{kj}}{\partial \theta}\Big\| \leq \bar{q}\xi\big(1 + \mathcal{M} + \frac{\bar{p}\,r(1 + \mathcal{M})}{1 - r}\big) = \bar{\mathcal{M}} < \infty, \tag{58}$$

which, by (21), implies $\frac{\partial \mathcal{L}_T}{\partial \theta}$ will be bounded for $T \to \infty$.

$(iii)$ Consider the PLRNN given by (27), where for simplicity we ignore the external inputs and noise terms. Let $\{\boldsymbol{z}_{t_1}, \boldsymbol{z}_{t_2}, \boldsymbol{z}_{t_3}, \ldots\}$ be an orbit which converges to $\Gamma_k$. Hence

$$\lim_{n \to \infty} d(\boldsymbol{z}_{t_n}, \Gamma_k) = 0, \tag{59}$$

which implies there exists a neighborhood $U$ of $\Gamma_k$ and $k$ sub-sequences $\{\boldsymbol{z}_{t_{km}}\}_{m=1}^{\infty}, \{\boldsymbol{z}_{t_{km+1}}\}_{m=1}^{\infty}$, $\cdots, \{\boldsymbol{z}_{t_{km+(k-1)}}\}_{m=1}^{\infty}$ of the sequence $\{\boldsymbol{z}_{t_n}\}_{n=1}^{\infty}$ such that all these sub-sequences belong to $U$ and

a) $\boldsymbol{z}_{t_{km+s}} = F^k(\boldsymbol{z}_{t_{k(m-1)+s}}), s = 0, 1, 2, \cdots, k-1,$

b) $\lim_{m \to \infty} \boldsymbol{z}_{t_{km+s}} = \boldsymbol{z}_{t^*k-s}, s = 0, 1, 2, \cdots, k-1,$

c) for every $\boldsymbol{z}_{t_n} \in U$ there is some $s \in \{0, 1, 2, \cdots, k-1\}$ such that $\boldsymbol{z}_{t_n} \in \{\boldsymbol{z}_{t_{km+s}}\}_{m=1}^{\infty}.$

In this case, for every $\boldsymbol{z}_{t_n} \in U$ with $\boldsymbol{z}_{t_n} \in \{\boldsymbol{z}_{t_{km+s}}\}_{m=1}^{\infty}$, there exists some $\tilde{n} \in \mathbb{N}$ such that $\boldsymbol{z}_{t_n} = \boldsymbol{z}_{t_{k\tilde{n}+s}}$ and $\lim_{\tilde{n} \to \infty} \boldsymbol{z}_{t_{k\tilde{n}+s}} = \boldsymbol{z}_{t^*k-s}$. Therefore, continuity of $F$ results in

$$\lim_{\tilde{n} \to \infty} F(\boldsymbol{z}_{t_{k\tilde{n}+s}}) = F(\boldsymbol{z}_{t^*k-s}), \tag{60}$$

and so by (28)

$$\lim_{\tilde{n} \to \infty} \left( \boldsymbol{W}_{\Omega(t_{k\tilde{n}+s})} \boldsymbol{z}_{t_{k\tilde{n}+s}} + \boldsymbol{h} \right) = \boldsymbol{W}_{\Omega(t^*k-s)} \boldsymbol{z}_{t^*k-s} + \boldsymbol{h}, \tag{61}$$

which implies

$$\lim_{\tilde{n} \to \infty} \boldsymbol{W}_{\Omega(t_{k\tilde{n}+s})} \boldsymbol{z}_{t_{k\tilde{n}+s}} = \boldsymbol{W}_{\Omega(t^*k-s)} \boldsymbol{z}_{t^*k-s}. \tag{62}$$

Assuming $\lim_{\tilde{n} \to \infty} \boldsymbol{W}_{\Omega(t_{k\tilde{n}+s})} = \boldsymbol{L}$, since (62) holds for every $\boldsymbol{z}_{t^*k-s}$, substituting $\boldsymbol{z}_{t^*k-s} = \boldsymbol{e}_1^{\mathsf{T}} = (1, 0, \cdots, 0)^T$ in (62), we can prove that the first column of $\boldsymbol{L}$ equals the first column of $\boldsymbol{W}_{\Omega(t^*k-s)}$. Performing the same procedure for $\boldsymbol{z}_{t^*k-s} = \boldsymbol{e}_i^{\mathsf{T}}, i = 2, 3, \cdots, M$, yields

$$\lim_{\tilde{n} \to \infty} \boldsymbol{W}_{\Omega(t_{k\tilde{n}+s})} = \boldsymbol{W}_{\Omega(t^*k-s)}. \tag{63}$$

According to (59), $U$ contains an infinite number of terms of the sequence $\{\boldsymbol{z}_{t_n}\}_{n=1}^{\infty}$, i.e.

$$\exists N \in \mathbb{N} \quad s.t. \quad n \geq N \implies \boldsymbol{z}_{t_n} \in U. \tag{64}$$

Suppose that $\boldsymbol{z}_{t_n} \in U$ for some $n \geq N$. Thus, there exists some $s \in \{0, 1, 2, \cdots, k-1\}$ such that $\boldsymbol{z}_{t_n} \in \{\boldsymbol{z}_{t_{km+s}}\}_{m=1}^{\infty}$. Without loss of generality let $s = 0$. Hence, there is some $\tilde{n} \in \mathbb{N}$ such that $\boldsymbol{z}_{t_n} = \boldsymbol{z}_{t_{k\tilde{n}}}$ and $\lim_{\tilde{n} \to \infty} \boldsymbol{z}_{t_{k\tilde{n}}} = \boldsymbol{z}_{t^*k}$. In this case, moving forward in time gives

$$\boldsymbol{z}_{t_n} = \boldsymbol{z}_{t_{k\tilde{n}}} \ \left( \boldsymbol{z}_{t_n} \in \{\boldsymbol{z}_{t_{km}}\}_{m=1}^{\infty} \right), \qquad\qquad \lim_{\tilde{n} \to \infty} \boldsymbol{z}_{t_{k\tilde{n}}} = \boldsymbol{z}_{t^*k},$$

$$\boldsymbol{z}_{t_{n+1}} = \boldsymbol{z}_{t_{k\tilde{n}+1}} \ \left( \boldsymbol{z}_{t_{n+1}} \in \{\boldsymbol{z}_{t_{km+1}}\}_{m=1}^{\infty} \right), \qquad\qquad \lim_{\tilde{n} \to \infty} \boldsymbol{z}_{t_{k\tilde{n}+1}} = \boldsymbol{z}_{t^*k-1},$$

$$\boldsymbol{z}_{t_{n+2}} = \boldsymbol{z}_{t_{k\tilde{n}+2}} \ \left( \boldsymbol{z}_{t_{n+2}} \in \{\boldsymbol{z}_{t_{km+2}}\}_{m=1}^{\infty} \right), \qquad\qquad \lim_{\tilde{n} \to \infty} \boldsymbol{z}_{t_{k\tilde{n}+2}} = \boldsymbol{z}_{t^*k-2},$$

$$\vdots$$

$$\boldsymbol{z}_{t_{n+k-1}} = \boldsymbol{z}_{t_{k\tilde{n}+k-1}} \ \left( \boldsymbol{z}_{t_{n+(k-1)}} \in \{\boldsymbol{z}_{t_{km+k-1}}\}_{m=1}^{\infty} \right), \qquad \lim_{\tilde{n} \to \infty} \boldsymbol{z}_{t_{k\tilde{n}+k-1}} = \boldsymbol{z}_{t^*k-(k-1)},$$

$$\boldsymbol{z}_{t_{n+k}} = \boldsymbol{z}_{t_{k(\tilde{n}+1)}} \ \left( \boldsymbol{z}_{t_{n+k}} \in \{\boldsymbol{z}_{t_{km}}\}_{m=1}^{\infty} \right), \qquad\qquad \lim_{\tilde{n} \to \infty} \boldsymbol{z}_{t_{k(\tilde{n}+1)}} = \boldsymbol{z}_{t^*k},$$

$$\boldsymbol{z}_{t_{n+k+1}} = \boldsymbol{z}_{t_{k(\tilde{n}+1)+1}} \ \left( \boldsymbol{z}_{t_{n+k+1}} \in \{\boldsymbol{z}_{t_{km+1}}\}_{m=1}^{\infty} \right), \qquad \lim_{\tilde{n} \to \infty} \boldsymbol{z}_{t_{k(\tilde{n}+1)+1}} = \boldsymbol{z}_{t^*k-1},$$

$$\vdots$$

$$\boldsymbol{z}_{t_{n+2k-1}} = \boldsymbol{z}_{t_{k(\tilde{n}+1)+k-1}} \ \left( \boldsymbol{z}_{t_{n+2k-1}} \in \{\boldsymbol{z}_{t_{km+k-1}}\}_{m=1}^{\infty} \right), \quad \lim_{\tilde{n} \to \infty} \boldsymbol{z}_{t_{k(\tilde{n}+1)+k-1}} = \boldsymbol{z}_{t^*k-(k-1)},$$

$$\boldsymbol{z}_{t_{n+2k}} = \boldsymbol{z}_{t_{k(\tilde{n}+2)}} \ \left( \boldsymbol{z}_{t_{n+2k}} \in \{\boldsymbol{z}_{t_{km}}\}_{m=1}^{\infty} \right), \qquad\qquad \lim_{\tilde{n} \to \infty} \boldsymbol{z}_{t_{k(\tilde{n}+2)}} = \boldsymbol{z}_{t^*k},$$

$$\vdots \tag{65}$$

Consequently, for $n \geq N$ and $j \in \mathbb{N}$, we can write

$$\prod_{i=0}^{kj-1} \boldsymbol{W}_{\Omega(t_{n+kj-1-i})}$$

$$= \Big( \prod_{i=1}^{k} \boldsymbol{W}_{\Omega(t_{k(\bar{n}+j)+k-i})} \Big) \Big( \prod_{i=1}^{k} \boldsymbol{W}_{\Omega(t_{k(\bar{n}+j-1)+k-i})} \Big) \cdots \Big( \prod_{i=1}^{k} \boldsymbol{W}_{\Omega(t_{k(\bar{n})+k-i})} \Big)$$

$$= \prod_{l=0}^{j} \prod_{i=1}^{k} \boldsymbol{W}_{\Omega(t_{k(\bar{n}+j-l)+k-i})}. \tag{66}$$

On the other hand, in equation (28), there are different configurations for matrix $\boldsymbol{D}_{\Omega(t-1)}$ and hence different forms for matrix $\boldsymbol{W}_{\Omega(t_{k\bar{n}+s})}$. In this case, the phase space of the system is divided into different sub-regions by some borders; see [63, 64] for more details. Also, since the system (28) is a linear map in each sub-region, the $k$ periodic points of $\Gamma_k$ must belong to different sub-regions (at least two different sub-regions). Accordingly, based on (63) and (65), there exists some $\tilde{N} \in \mathbb{N}$ such that for every $\tilde{n} \geq \tilde{N}$ both $\boldsymbol{z}_{t_{k\bar{n}+s}}$ and $\boldsymbol{z}_{t^{*k}-s}$ belong to the same sub-region and so the matrices $\boldsymbol{W}_{\Omega(t_{k\bar{n}+s})}$ and $\boldsymbol{W}_{\Omega(t^{*k}-s)}$ ($s \in \{0, 1, 2, \cdots, k-1\}$) are identical. Hence, for $n \geq N$, $\tilde{n} \geq \tilde{N}$ and $j \in \mathbb{N}$, equation (66) becomes

$$\prod_{i=0}^{kj-1} \boldsymbol{W}_{\Omega(t_{n+kj-1-i})} = \prod_{l=0}^{j} \prod_{i=1}^{k} \boldsymbol{W}_{\Omega(t_{k(\bar{n}+j-l)+k-i})} = \Big( \prod_{s=0}^{k-1} \boldsymbol{W}_{\Omega(t^{*k}-s)} \Big)^{j}. \tag{67}$$

Therefore, similar to the part $(ii)$, we can prove for every $\boldsymbol{z}_1 \in \mathcal{B}_{\Gamma_k}$, $\frac{\partial \boldsymbol{z}_T}{\partial \theta}$ and $\frac{\partial \mathcal{L}_T}{\partial \theta}$ will also remain bounded. □

### A.2.2 Proof of theorem 2, part (ii)

*Proof.* $(ii)$ Let for every $T > 2$

$$\boldsymbol{L}_T := J_T^* J_{T-1}^* \cdots J_2^*. \tag{68}$$

$\{\boldsymbol{L}_T\}_{T \in \mathbb{N}, T>2}$ is a sequence of matrices $\boldsymbol{L}_T = [l_{ij}^{(T)}]_{1 \leq i,j \leq M}$ and, due to (13), $\lim_{T \to \infty} \|\boldsymbol{L}_T\| = \infty$. Hence, there is at least one sub-sequence $\{l_{mk}^{(T_n)}\}_{T_n \in \mathbb{N}, T_n>2}$ (for some $m, k \in \{1, 2, \cdots, M\}$) such that $\lim_{T_n \to \infty} l_{mk}^{(T_n)} = \infty$.

On the other hand

$$\frac{\partial \boldsymbol{z}_T^*}{\partial \theta} = \frac{\partial^+ \boldsymbol{z}_T^*}{\partial \theta} + \sum_{t=1}^{T-2} \Big( \prod_{r=0}^{t-1} J_{T-r}^* \Big) \frac{\partial^+ \boldsymbol{z}_{T-t}^*}{\partial \theta}. \tag{69}$$

Moreover, there exists some $N > 2$ such that (for $t = T - N + 1$)

$$\frac{\partial^+ \boldsymbol{z}_{N-1}^*}{\partial \theta} \neq 0. \tag{70}$$

For $\theta$ as the $k$-th element of a parameter vector $\boldsymbol{\theta}$ (or belonging to the $k$-th row of a parameter matrix $\boldsymbol{\theta}$), the term

$$\Big( \prod_{r=0}^{T-N} J_{T-r}^* \Big) \frac{\partial^+ \boldsymbol{z}_{N-1}^*}{\partial \theta} \tag{71}$$

is a vector in which the $i$-th element is $l_{ik}^{(T)} \frac{\partial^+ z_{k,N-1}^*}{\partial \theta}$.

Since $\lim_{T_n \to \infty} l_{mk}^{(T_n)} = \infty$, due to (70) $\lim_{T_n \to \infty} l_{mk}^{(T_n)} \frac{\partial^+ z_{k,N-1}^*}{\partial \theta} = \infty$, which implies $\frac{\partial \boldsymbol{z}_T^*}{\partial \theta}$ will diverge as $T \to \infty$. Similarly, by (21), we can prove $\frac{\partial \mathcal{L}_T^*}{\partial \theta}$ is divergent for $T \to \infty$.

By Oseledec's multiplicative ergodic Theorem, the results also hold for every $\boldsymbol{z}_1 \in \mathcal{B}_{\Gamma^*}$. □

### A.2.3 Proof of theorem 3

*Proof.* Let $\Gamma = \{z_1, z_2, \ldots z_T, \cdots\}$ be a quasi-periodic attractor. Then, the largest Lyapunov exponent of $\Gamma$ is

$$\lambda = \lim_{T \to \infty} \frac{1}{T} \ln \|J_T J_{T-1} \cdots J_2\| = \lim_{T \to \infty} \frac{1}{T} \ln \left\| \frac{\partial z_T}{\partial z_1} \right\| = 0. \tag{72}$$

We prove for every $0 < \epsilon < 1$

$$\lim_{T \to \infty} (1 - \epsilon)^{T-1} < \lim_{T \to \infty} \left\| \frac{\partial z_T}{\partial z_1} \right\| < \lim_{T \to \infty} (1 + \epsilon)^{T-1}. \tag{73}$$

For this purpose, we show $\forall \, 0 < \epsilon < 1$

(I) $\lim_{T \to \infty} (1 - \epsilon)^{T-1} < \lim_{T \to \infty} \left\| \frac{\partial z_T}{\partial z_1} \right\|$, and

(II) $\lim_{T \to \infty} \left\| \frac{\partial z_T}{\partial z_1} \right\| < \lim_{T \to \infty} (1 + \epsilon)^{T-1}$.

Assume for the sake of contradiction that (I) does not hold. Then there exists some $0 < \epsilon < 1$ such that

$$\lim_{T \to \infty} (1 - \epsilon)^{T-1} \geq \lim_{T \to \infty} \left\| \frac{\partial z_T}{\partial z_1} \right\|. \tag{74}$$

Therefore

$$\exists \, T_0 > 1 \;\; s.t. \;\; \forall \, T \geq T_0 \implies (1 - \epsilon)^{T-1} \geq \left\| \frac{\partial z_T}{\partial z_1} \right\|, \tag{75}$$

and so

$$\exists \, T_0 > 1 \;\; s.t. \;\; \forall \, T \geq T_0 \implies \frac{\ln(1 - \epsilon)^{T-1}}{T - 1} \geq \frac{\ln \left\| \frac{\partial z_T}{\partial z_1} \right\|}{T - 1}. \tag{76}$$

Consequently, due to (72), for $T \to \infty$ we have $\ln(1 - \epsilon) \geq 0$. This implies $\epsilon \leq 0$, which is a contradiction.

Similarly if we assume (II) is not true, then there exists some $0 < \epsilon < 1$ such that

$$\lim_{T \to \infty} \left\| \frac{\partial z_T}{\partial z_1} \right\| \geq \lim_{T \to \infty} (1 + \epsilon)^{T-1}. \tag{77}$$

Thereby

$$\exists \, T_0 > 1 \;\; s.t. \;\; \forall \, T \geq T_0 \implies \left\| \frac{\partial z_T}{\partial z_1} \right\| \geq (1 + \epsilon)^{T-1}, \tag{78}$$

and thus

$$\exists \, T_0 > 1 \;\; s.t. \;\; \forall \, T \geq T_0 \implies \frac{\ln \left\| \frac{\partial z_T}{\partial z_1} \right\|}{T - 1} \geq \frac{\ln(1 + \epsilon)^{T-1}}{T - 1}. \tag{79}$$

This means $\ln(1 + \epsilon) \leq 0$ as $T \to \infty$, i.e. $\epsilon \leq 0$, which is a contradiction.

Therefore (14) holds for $\Gamma$ and also, according to Oseledec's multiplicative ergodic Theorem, for every $z_1$ in the basin of attraction of $\Gamma$. $\qquad \square$

### A.3 Additional results on relation between dynamics and gradients

#### A.3.1 Further results and remarks related to Theorem 2

**Remark 4.** *The result of Theorem 2 also holds for unstable orbits $\{z_1, z_2, z_3, \cdots\}$ with positive largest Lyapunov exponent. Trivially, for such orbits that diverge to infinity (unbounded latent states) gradients of the loss function will explode as $T \to \infty$.*

**Remark 5.** *For RNNs with ReLU activation functions there are finite compartments in the phase space each with a different functional form. In such a case, to define the largest Lyapunov exponent of $\Gamma^*$, in the proof of Theorem 2 we assume that $\Gamma^*$ never maps to the points of the borders.*

Based on Theorem 2, we can also formulate the necessary conditions for chaos and diverging gradients in standard RNNs with particular activation functions by considering the norms of their recurrence matrix, for which the following Corollary provides the basis:

**Corollary 1.** *Let for a standard RNN*

$$\left\| diag\big(f'\big(\boldsymbol{W}\boldsymbol{z}_{t-1} + \boldsymbol{B}\boldsymbol{s}_t + \boldsymbol{h}\big)\big) \right\| \le \gamma < \infty. \tag{80}$$

*If the RNN is chaotic, then $\|\boldsymbol{W}\| \, \gamma > 1$.*

*Proof.* Assume for the sake of contradiction that $\|\boldsymbol{W}\| \, \gamma \le 1$. From

$$\left\| \prod_{2 < t \le T} \boldsymbol{W} diag\big(f'\big(\boldsymbol{W}\boldsymbol{z}_{t-1} + \boldsymbol{B}\boldsymbol{s}_t + \boldsymbol{h}\big)\big) \right\| \le \prod_{2 < t \le T} \left\| \boldsymbol{W} diag\big(f'\big(\boldsymbol{W}\boldsymbol{z}_{t-1} + \boldsymbol{B}\boldsymbol{s}_t + \boldsymbol{h}\big)\big) \right\|$$

$$\le (\|\boldsymbol{W}\| \, \gamma)^{T-2}, \tag{81}$$

it is concluded that $\lim_{T \to \infty} \left\| \prod_{2 < t \le T} \boldsymbol{W} diag\big(f'\big(\boldsymbol{W}\boldsymbol{z}_{t-1} + \boldsymbol{B}\boldsymbol{s}_t + \boldsymbol{h}\big)\big) \right\| < \infty$, which contradicts (13). This means $\|\boldsymbol{W}\| \, \gamma > 1$ is a necessary condition for the standard RNN to be chaotic. $\qquad\square$

**Remark 6.** *For RNN with the tanh and sigmoid activation functions $\gamma = 1$ and $\gamma = \frac{1}{4}$, respectively. Thus, by Corollary 1, the necessary conditions for chaos in these two cases are $\|\boldsymbol{W}\| > 1$ and $\|\boldsymbol{W}\| > 4$, respectively.*

#### A.3.2 Other connections between dynamics and gradients

There is a direct link between the norms of the Jacobians of the RNN along trajectories and the EVGP. By observing this link, we can formulate some general conditions that will have implications for the behavior of the gradients regardless of the limiting behavior of the RNN, as collected in the following theorem:

**Theorem 4.** *Let $\mathcal{O}_{\boldsymbol{z}_1} = \{\boldsymbol{z}_1, \boldsymbol{z}_2, \ldots \boldsymbol{z}_T, \cdots\}$ be a sequence (orbit) generated by an RNN $F_{\boldsymbol{\theta}} \in \mathcal{R}$ parameterized by $\boldsymbol{\theta}$, and $\boldsymbol{P}_T := \boldsymbol{J}_T - \boldsymbol{I}$, $T = 2, 3, \cdots$.*

*(i) Assume that $\mathcal{O}_{\boldsymbol{z}_1}$ is an orbit for which $\left\| \frac{\partial^+ \boldsymbol{z}_T}{\partial \boldsymbol{\theta}} \right\| \le \xi \ \forall t$. If $\sum_{T=2}^{\infty} \|\boldsymbol{J}_T\| < \infty$, then the Jacobian $\frac{\partial \boldsymbol{z}_T}{\partial \boldsymbol{z}_1}$, the tangent vector $\frac{\partial \boldsymbol{z}_T}{\partial \boldsymbol{\theta}}$ and thus the gradient of the loss function, $\frac{\partial \mathcal{L}_T}{\partial \boldsymbol{\theta}}$, will be bounded for $T \to \infty$.*

*(ii) If $\sum_{T=2}^{\infty} \|\boldsymbol{P}_T\| < \infty$, then the Jacobian $\frac{\partial \boldsymbol{z}_T}{\partial \boldsymbol{z}_1}$ will neither vanish nor explode as $T \to \infty$.*

*(iii) Let $\|\boldsymbol{J}_T\| \ne 0$, $T \ge 2$, and $\sum_{T=2}^{\infty} \ln \|\boldsymbol{J}_T\|$ diverge to $-\infty$, then the Jacobian $\frac{\partial \boldsymbol{z}_T}{\partial \boldsymbol{z}_1}$ vanishes as $T$ tends to infinity.*

Part $(i)$ of Theorem 4 relaxes some of the conditions required in Theorem 1 for bounded gradients by imposing a Lipschitz condition on the immediate derivatives. Part $(ii)$ generalizes conditions satisfied, for instance, in orthogonal (unitary) RNNs [4, 32] or fully regularized PLRNNs [80].

*Proof.* Let $\|.\|$ be any matrix norm satisfying $\|\boldsymbol{A}_1 \boldsymbol{A}_2\| \le \|\boldsymbol{A}_1\| \|\boldsymbol{A}_2\|$.

$(i)$ By boundedness of $\frac{\partial^+ \boldsymbol{z}_T}{\partial \theta}$ we have

$$\left\| \frac{\partial \boldsymbol{z}_T}{\partial \theta} \right\| = \left\| \frac{\partial^+ \boldsymbol{z}_T}{\partial \theta} + \sum_{t=1}^{T-2} \left( \prod_{r=0}^{t-1} \boldsymbol{J}_{T-r} \right) \frac{\partial^+ \boldsymbol{z}_{T-t}}{\partial \theta} \right\|$$

$$\leq \xi \left( 1 + \sum_{t=1}^{T-2} \left\| \prod_{r=0}^{t-1} \boldsymbol{J}_{T-r} \right\| \right) \leq \xi \left( 1 + \sum_{t=1}^{T-2} \prod_{r=0}^{t-1} \| \boldsymbol{J}_{T-r} \| \right). \tag{82}$$

Moreover,

$$1 + \sum_{t=1}^{T-2} \prod_{r=0}^{t-1} \| \boldsymbol{J}_{T-r} \| \leq 1 + \sum_p \| \boldsymbol{J}_p \| + \sum_{p<q} \| \boldsymbol{J}_p \| \, \| \boldsymbol{J}_q \| + \sum_{p<q<r} \| \boldsymbol{J}_p \| \, \| \boldsymbol{J}_q \| \, \| \boldsymbol{J}_r \| + \cdots$$

$$= \left( 1 + \| \boldsymbol{J}_T \| \right) \left( 1 + \| \boldsymbol{J}_{T-1} \| \right) \cdots \left( 1 + \| \boldsymbol{J}_2 \| \right) =: \prod_{t=2}^{T} \left( 1 + \| \boldsymbol{J}_t \| \right). \tag{83}$$

Since $\sum_{T=2}^{\infty} \| \boldsymbol{J}_T \|$ converges, according to [94], the infinite products $\prod_{T=2}^{\infty} \left( 1 + \| \boldsymbol{J}_T \| \right)$ in (83) converge to a finite number $\tilde{\mathcal{K}} \neq 0$. Consequently, by (82) and (83)

$$\lim_{T \to \infty} \left\| \frac{\partial \boldsymbol{z}_T}{\partial \theta} \right\| \leq \tilde{\mathcal{K}} < \infty, \tag{84}$$

which implies $\frac{\partial \mathcal{L}_T}{\partial \theta}$ will be bounded for $T \to \infty$.

Furthermore

$$\lim_{T \to \infty} \left\| \frac{\partial \boldsymbol{z}_T}{\partial \boldsymbol{z}_1} \right\| \leq \prod_{T=2}^{\infty} \| \boldsymbol{J}_T \| := \lim_{T \to \infty} \left( \| \boldsymbol{J}_T \| \, \| \boldsymbol{J}_{T-1} \| \cdots \| \boldsymbol{J}_2 \| \right) \leq \prod_{T=2}^{\infty} \left( 1 + \| \boldsymbol{J}_T \| \right) \leq \tilde{\mathcal{K}}, \tag{85}$$

which completes the proof.

$(ii)$ Since $\sum_{T=1}^{\infty} \| \boldsymbol{P}_T \| < \infty$, due to [94] the infinite product

$$\prod_{T=2}^{\infty} \left( \boldsymbol{I} + \boldsymbol{P}_T \right) = \prod_{T=2}^{\infty} \boldsymbol{J}_T := \lim_{T \to \infty} \boldsymbol{J}_T \, \boldsymbol{J}_{T-1} \cdots \boldsymbol{J}_2, \tag{86}$$

converges to a matrix $\boldsymbol{K} \neq \boldsymbol{O}$, which implies

$$0 < \lim_{T \to \infty} \left\| \frac{\partial \boldsymbol{z}_T}{\partial \boldsymbol{z}_1} \right\| = \| \boldsymbol{K} \| < \infty. \tag{87}$$

$(iii)$ For $\| \boldsymbol{J}_T \| \neq 0, \; T \geq 2$, we have

$$0 \leq \left\| \frac{\partial \boldsymbol{z}_T}{\partial \boldsymbol{z}_1} \right\| \leq \| \boldsymbol{J}_T \| \, \| \boldsymbol{J}_{T-1} \| \cdots \| \boldsymbol{J}_2 \|$$

$$= e^{\ln \| \boldsymbol{J}_T \|} e^{\ln \| \boldsymbol{J}_{T-1} \|} \cdots e^{\ln \| \boldsymbol{J}_2 \|} = e^{\sum_{t=2}^{T} \ln \| \boldsymbol{J}_t \|}. \tag{88}$$

Hence if $\sum_{T=2}^{\infty} \ln \| \boldsymbol{J}_T \| \to -\infty$, then

$$\lim_{T \to \infty} \frac{\partial \boldsymbol{z}_T}{\partial \boldsymbol{z}_1} = \boldsymbol{O}. \tag{89}$$

$\square$

## A.4 Empirical evaluation: Datasets

**Lorenz attractor** The Lorenz system [56] is a simplified model for atmospheric convection, given by

$$
\begin{aligned}
\frac{\mathrm{d}x}{\mathrm{d}t} &= \sigma(y - x), \\
\frac{\mathrm{d}y}{\mathrm{d}t} &= x(\rho - z) - y, \\
\frac{\mathrm{d}z}{\mathrm{d}t} &= xy - \beta z.
\end{aligned}
\tag{90}
$$

The system is of particular interest for its chaotic regime and was studied here for $\sigma = 16$, $\rho = 45.92$ and $\beta = 4$. For these parameters the Lorenz system is known to have a maximal Lyapunov exponent $\lambda_{\mathrm{max}} = 1.5$ [72]. To generate a time series, the ODEs were integrated with a step size $\Delta t = 0.01$ using `scipy.integrate`. Accordingly, the prediction time is $\tau_{pred} = \frac{\ln(2)}{\Delta t\, \lambda_{max}} = 46.2$.

**Duffing oscillator** The Duffing oscillator [15] is an example of a periodically forced oscillator with nonlinear elasticity

$$
\ddot{x} + \delta\dot{x} + \beta x + \alpha x^3 = \gamma\cos(\omega t).
\tag{91}
$$

Note that this system is non-autonomous, that is externally forced due to the r.h.s. of eqn. 91. The following parameters were chosen to arrive at a chaotically forced oscillator: $\alpha = 1.0$, $\beta = -1.0$, $\delta = 0.1$, $\gamma = 0.35$, and $\omega = 1.4$. For these parameters the Duffing oscillator has a maximum Lyapunov exponent of $\lambda_{max} = 0.0995$. The dataset used here was created with the code from [24] as a three dimensional embedding with step size $\Delta t = 0.17$. The prediction time is $\tau_{pred} = 39.28$.

**Rössler system** Another prime textbook example for a chaotic system is the Rössler system [76] given by:

$$
\begin{aligned}
\frac{dx}{dt} &= -y - z, \\
\frac{dy}{dt} &= x + ay, \\
\frac{dz}{dt} &= b + z(x - c).
\end{aligned}
\tag{92}
$$

For the parameters $a = 0.15$, $b = 0.2$ and $c = 10$, the maximal Lyapunov exponent is $\lambda_{\mathrm{max}} = 0.09$ [72]. To arrive at a time series, a step size of $\Delta t = 0.1$ was chosen for integration. This gives us a prediction time of $\tau_{pred} = 77.0$ for this system.

**Mackey-Glass equation** The Mackey-Glass equation [25] is a nonlinear time delay differential equation

$$
\dot{x} = \beta\frac{x_\rho}{1 + x_\rho^n} - \gamma x \quad \text{with } \beta, \gamma, \rho > 0.
\tag{93}
$$

Here $x_\rho$ represents the value of the variable $x$ at time $t - \rho$ (note that strictly, mathematically, this makes the system infinite-dimensional). Choosing the parameters to be $\beta = 2$, $\gamma = 1.0$, $n = 9.65$, and $\rho = 2.0$, leads to chaotic behavior with a maximum Lyapunov exponent of $\lambda_{max} = 0.21$. The dataset was created as a 10-dimensional embedding with the code from [24] using $\Delta t = 0.04$. This yields a prediction time of $\tau_{pred} = 82.2$.

**Empirical temperature time series** This time series was recorded at the Weather Station at the Max Planck Institute for Biogeochemistry in Jena, Germany, spanning the time period between 2009 and 2016, and reassembled by François Chollet for the book *Deep Learning with Python*. The data set can be accessed at https://www.kaggle.com/pankrzysiu/weather-archive-jena.
To expose the underlying chaotic dynamics of the time series, trends and yearly cycles were removed, and nonlinear noise-reduction was performed (using `ghkss` from *TISEAN*, see also [43]). Fig. 5 (a) shows a snippet of the temperature data in comparison with the de-noised time-series. High-frequency

noise was further reduced through Gaussian kernel smoothing ($\sigma = 200$), and the resulting time series was sub-sampled (every $5^{th}$ data point was retained). Fig. 5 (b) clearly reveals a fractional dimension of $D_{eff} = 2.8$ for the de-noised and smoothed time-series. This strongly suggests that the dynamics governing the time series are chaotic. We created a time delay embedding [42] with $m = 5$ (estimated by the false nearest neighbor technique, see [44]) and delay $\Delta t = 500$ (obtained as the first minimum of the mutual information). The first three embedding dimensions are shown in Fig. 5(c). The maximal Lyapunov exponent of this time series was determined with `lyap_r` from *TISEAN* [30] to be $\lambda_{\max} = 0.016$, see Fig. 4(a). This value is in close agreement with the literature [62]. The predictability time of this system is estimated to be $\tau_{pred} = 43.3$.

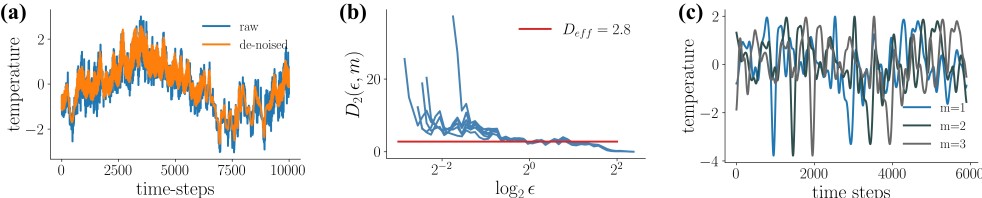

Figure 5: (a) Snippet of the original temperature data and de-noised time series. (b) Blue lines show the local slopes of the correlation sums for embedding dimensions $m \in \{5, \dots, 10\}$. The convergence of these estimates in $m$ reveals a fractional dimension indicated by the plateau. (c) First three dimensions of the time-delay embedding series as used for training.

All datasets used were standardized (i.e., centered with unit variance) prior to training.

## A.5 Empirical evaluation: measures of reconstruction quality

**Attractor overlap** To asses the geometrical similarity of the chaotic attractor produced by the RNN to the one underlying the observations, we calculate the Kullback-Leibler divergence of the ground truth distribution $p_{\text{true}}(\boldsymbol{x})$ and the distribution $p_{\text{gen}}(\boldsymbol{x}|\boldsymbol{z})$ generated by RNN simulation. To do so in practice, we employ a binning approximation (see [50])

$$D_{\text{stsp}}\left(p_{\text{true}}(\mathbf{x}), p_{\text{gen}}(\mathbf{x} \mid \mathbf{z})\right) \approx \sum_{k=1}^{K} \hat{p}_{\text{true}}^{(k)}(\mathbf{x}) \log\left(\frac{\hat{p}_{\text{true}}^{(k)}(\mathbf{x})}{\hat{p}_{\text{gen}}^{(k)}(\mathbf{x} \mid \mathbf{z})}\right),$$

where $K$ is the total number of bins, and $\hat{p}_{\text{true}}^{(k)}(\mathbf{x})$ and $\hat{p}_{\text{gen}}^{(k)}(\mathbf{x} \mid \mathbf{z})$ are estimates obtained as relative frequencies through sampling trajectories from the observed time-series and the trained RNN, respectively.

**Hellinger distance between power spectra** Since in DS reconstruction we mainly aim to capture invariant and time-independent properties of the underlying system, besides the geometrical agreement, we compare the similarity in true and RNN-reconstructed power spectra. To do so, we generate a time series of length $100,000$ from the RNN and calculate its dimension-wise power spectra $S(\omega)$ using the fast Fourier transform (`scipy.fft`). By standardizing all trajectories prior to Fourier transforming them, we have $\int_{-\infty}^{\infty} S(\omega) = 1$ due to the Plancherel theorem. This allows us to compare two power spectra, $S(\omega)$ and $P(\omega)$, with the Hellinger distance

$$H(S(\omega), P(\omega)) = \sqrt{1 - \int_{-\infty}^{\infty} \sqrt{S(\omega)P(\omega)}\, d\omega} \quad \in [0, 1]. \tag{94}$$

To reduce the influence of noise we apply Gaussian kernel smoothing. The Hellinger distances between observed and generated spectra for all dimensions are then averaged to give the reported overall distance $D_H$.

## A.6 Further empirical evaluations

### A.6.1 Reconstruction: Rössler System

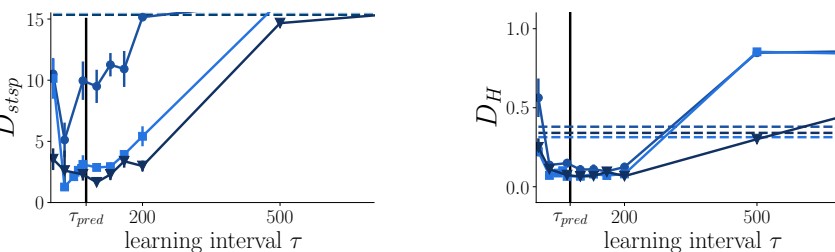

Figure 6: Overlap in attractor geometry ($D_{stsp}$, lower = better) and dimension-wise comparison of power-spectra ($D_H$, lower = better) against learning interval $\tau$ for the Rössler attractor. Continuous lines = sparsely forced BPTT. Dashed lines = classical BPTT with gradient clipping. Prediction time indicated vertically in black.

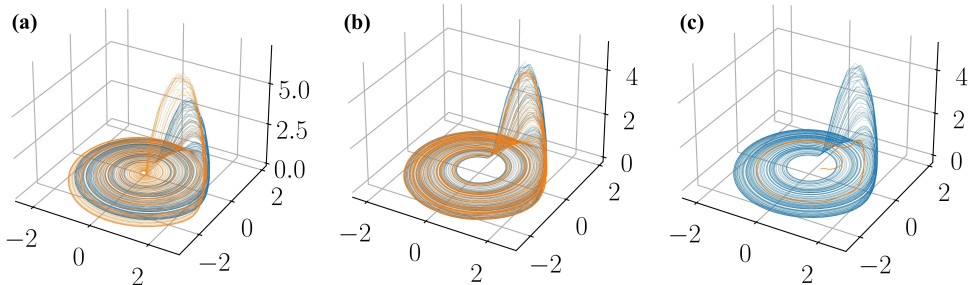

Figure 7: The Rössler attractor (blue) and reconstruction by a LSTM (orange) trained with a learning interval (a) chosen too small ($\tau = 5$), (b) chosen optimally ($\tau = 30$), and (c) chosen too large ($\tau = 200$).

### A.6.2 Reconstruction: High-dimensional Mackey-Glass system

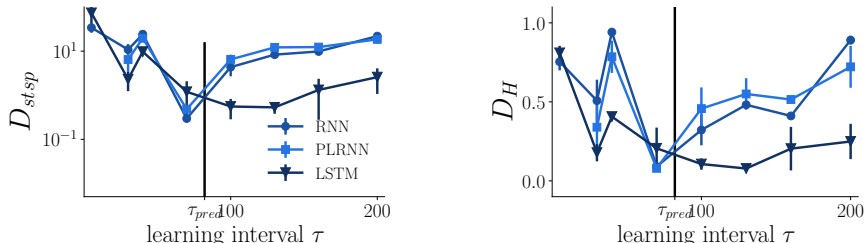

Figure 8: Overlap in attractor geometry ($D_{stsp}$, lower = better) and dimension-wise comparison of power-spectra ($D_H$, lower = better) against learning interval $\tau$ for the 10d Mackey-Glass system. Continuous lines = sparsely forced BPTT. Prediction time indicated vertically in black.

### A.6.3 Reconstruction: Partially observed Lorenz System

For this evaluation we trained models only on the variables $\{y, z\}$ of the Lorenz system, eqn. 90. In order to compute the attractor overlap ($D_{stsp}$) in the true state space, however, after training the observation matrix $B$ was recomputed by linearly regressing the first 10 latent states onto the first 10 observations from all three Lorenz variables in eqn. 90.

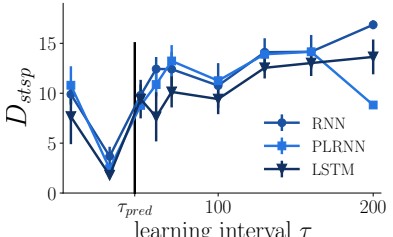 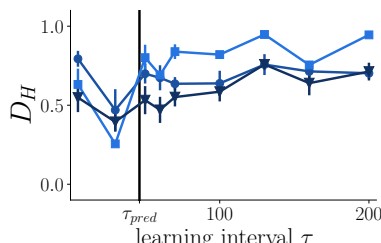

Figure 9: Overlap in attractor geometry ($D_{stsp}$, lower = better) and dimension-wise comparison of power-spectra ($D_H$, lower = better) against learning interval $\tau$ for the partially observed Lorenz system. Continuous lines = sparsely forced BPTT. Prediction time indicated vertically in black.

### A.6.4 Other initialization procedures: Truncated BPTT with zero resetting or forward-iterated states

A common procedure in training RNNs is partitioning the time series into chunks of length $\tau$ (as we did based on the Lyapunov spectrum), but then simply resetting the hidden states $\mathbf{z}_{1(k)}$ at the beginning of each chunk (window) $k$ to $\mathbf{0}$, or forward-iterating them from the previous chunk $k - 1$, i.e. $\mathbf{z}_{1(k)} = F_{\boldsymbol{\theta}}(\mathbf{z}_{\tau(k-1)})$. Formally this would mean that we do not force the trajectory back on track as in our approach, but instead may either kick it off track (zero resetting) or just let it freely evolve whilst still truncating the gradients (forward-iterating). To illustrate this, here we trained an LSTM on chunks (windows) with a length given by the optimal $\tau$ ($\tau_{opt} = 30$ for the Lorenz system), but then initialized the hidden states to either 0 or to the forward-iterated last state at the beginning of each window. The performance obtained with zero-resetting is indicated by the dashed line in Fig. 10a below, while the performance with forward-iterated states is shown in Fig. 10c. As another control, we also checked dependence on window length (without forcing) in Fig. 10b.

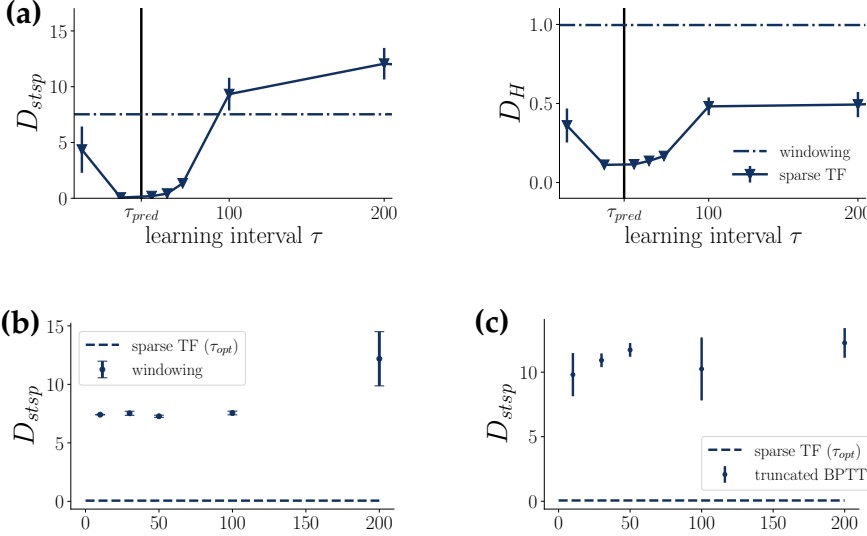

Figure 10: (a) Overlap in attractor geometry ($D_{stsp}$, lower = better) and dimension-wise comparison of power-spectra ($D_H$, lower = better) against learning interval $\tau$ for the Lorenz system. Continuous lines = sparsely forced BPTT. Dashed-dotted lines = windowing without forcing (choosing windows according to the optimal prediction time, but resetting hidden states to zero rather than its TF control value). Prediction time indicated vertically in black. (b) Dependence of geometrical reconstruction quality on window length. Without forcing, the window length hardly has any bearing on reconstruction quality. (c) Same as (b) but with initial states of each window $k$ forward-iterated from the previous window's state, $\mathbf{z}_{1(k)} = F_{\boldsymbol{\theta}}(\mathbf{z}_{\tau(k-1)})$, instead of zero resetting.

### A.6.5 Electroencephalogram (EEG) data

We used EEG data recorded by Schalk et al. [79] and provided on PhysioNet [27], from which we took the baseline recording of the first patient for our analysis. Preprocessing was performed as outlined above for the temperature time-series, i.e. we applied nonlinear noise-reduction (see Fig. 11 (a)) and Gaussian kernel smoothing ($\sigma = 5$). Fig. 11 (b) indicates a fractional dimension $D_{eff} = 2.5$ for the de-noised and smoothed times series. We created a time delay embedding with an embedding dimension of $m = 10$ and a delay time of $\Delta t = 40$. The maximal Lyapunov exponent for this time series was determined to be $\lambda_{max} = 0.017$, see Fig. 12 (a). With this, we obtain a predictability time $\tau_{pred} = 40.77$.

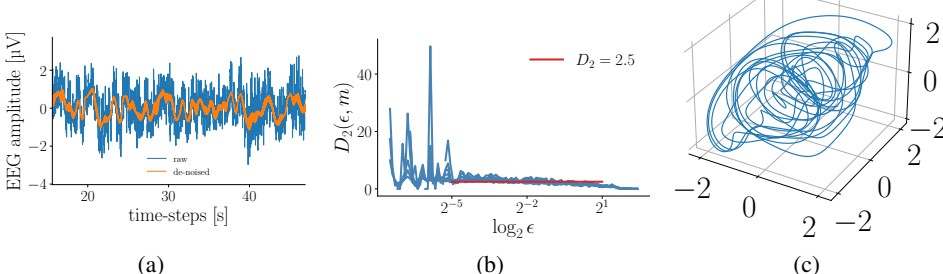

Figure 11: (a) Snippet of the original EEG data and de-noised time series. (b) Blue lines show the local slopes of the correlation sums for embedding dimensions $m \in \{5, \ldots, 15\}$. The convergence of these estimates in $m$ reveals a fractional dimension indicated by the plateau. (c) First three dimensions of the time-delay embedding series as used for training.

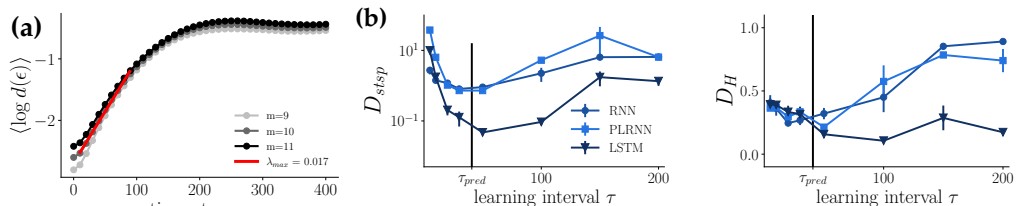

Figure 12: (a) The maximal Lyapunov exponent was determined as the slope of the average log-divergence of nearest neighbors in embedding space ($m$ = embedding dimension). (b) Reconstruction quality assessed by attractor overlap (lower = better) and dimension-wise comparison of power-spectra ($D_H$, lower = better). Black vertical lines = $\tau_{pred}$.

### A.6.6 Miscellaneous additional results

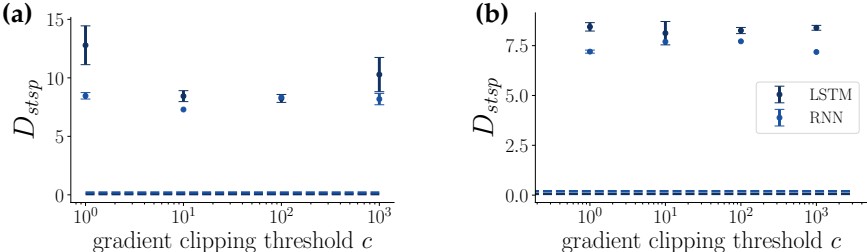

Figure 13: Dependence of geometrical reconstruction quality ($D_{stsp}$) on the Lorenz system for various clipping thresholds in classical BPTT. (a) Gradient clipping by constraining the Euclidean norm to $c$. (b) Gradient clipping by constraining the max (infinity) norm to $c$. For comparison, in both graphs the values obtained for sparse teacher forcing with optimal forcing interval $\tau_{pred}$ are shown as dashed lines.

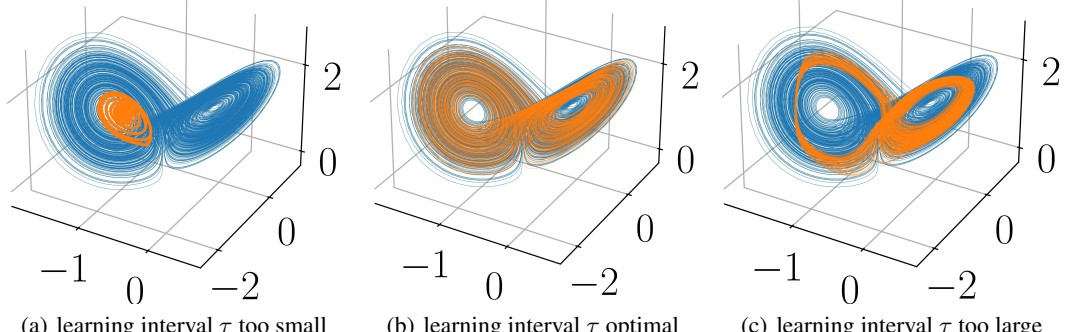

(a) learning interval $\tau$ too small     (b) learning interval $\tau$ optimal     (c) learning interval $\tau$ too large

Figure 14: Same as Fig. 3 for vanilla RNNs. Although, as this graph confirms, with sparsely forced BPTT training of vanilla RNNs on chaotic systems becomes feasible, generally they were somewhat harder to train than the other RNN architectures (likely due to their known problems with long-range dependencies).

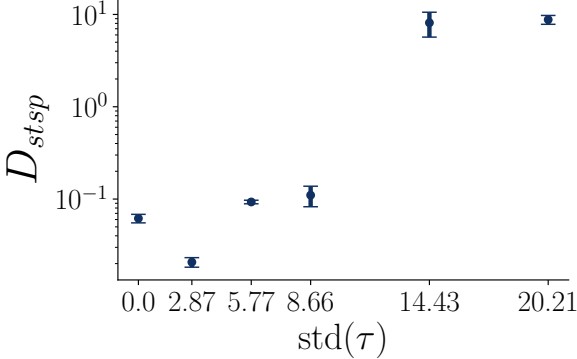

Figure 15: Teacher forcing for LSTM with the learning interval $\tau$ drawn uniformly random around the optimal value ($\tau_{opt} = 30$) with standard deviation $\text{std}(\tau)$ (for $\text{std}(\tau) > 8.66$ the interval becomes asymmetric, however, due to the lower bound at $\tau = 1$). As $\text{std}(\tau)$ is increased, performance generally degrades. A little jittering around the optimal interval $\tau_{pred}$ may potentially help, however (as more commonly observed in various machine learning procedures).

### A.7 Sparsely forced BPTT

**Loss truncation** One implicit consequence of the teacher forcing, eqn. (16), is the interruption of the hidden-to-hidden connections at these time points. More specifically, if the system is forced at time $t \in \mathcal{T}$, then there is no connection between $z_t$ and $z_{t+1}$, that is

$$\boldsymbol{J}_{t+1} = \frac{\partial \boldsymbol{z}_{t+1}}{\partial \boldsymbol{z}_t} = \frac{\partial RNN(\tilde{\boldsymbol{z}}_t)}{\partial \boldsymbol{z}_t} = 0. \tag{95}$$

To see how these vanishing Jacobians truncate the loss gradients w.r.t to some parameter $\theta$, let us focus on the loss gradients immediately after the forcing,

$$\begin{aligned}
\frac{\partial \mathcal{L}_{t+1}}{\partial \theta} &= \frac{\partial \mathcal{L}_{t+1}}{\partial \boldsymbol{z}_{t+1}} \sum_{k=1}^{t+1} \frac{\partial \boldsymbol{z}_{t+1}}{\partial \boldsymbol{z}_k} \frac{\partial^+ \boldsymbol{z}_k}{\partial \theta} \\
&= \frac{\partial \mathcal{L}_{t+1}}{\partial \boldsymbol{z}_{t+1}} \Big( \frac{\partial^+ \boldsymbol{z}_{t+1}}{\partial \theta} + \sum_{k=1}^{t} \underbrace{\frac{\partial \boldsymbol{z}_{t+1}}{\partial \boldsymbol{z}_k}}_{=0 \text{ , because of (95)}} \frac{\partial^+ \boldsymbol{z}_k}{\partial \theta} \Big) \\
&= \frac{\partial \mathcal{L}_{t+1}}{\partial \boldsymbol{z}_{t+1}} \frac{\partial^+ \boldsymbol{z}_{t+1}}{\partial \theta}.
\end{aligned} \tag{96}$$

Eqn. (96) shows that sparsely forced BPTT implicitly truncates the loss gradients because it interrupts the hidden-to-hidden connection from $z_t$ to $z_{t+1}$ for $t \in \mathcal{T}$. More generally, defining $\widetilde{t} := \max\{t' \in$

$\mathcal{T} : t' \le t\}$, the overall loss gradients are truncated to

$$\frac{\partial \mathcal{L}}{\partial \theta} = \sum_{t=1}^{T} \frac{\partial \mathcal{L}_t}{\partial z_t} \sum_{k=1}^{t} \frac{\partial z_t}{\partial z_k} \frac{\partial^+ z_k}{\partial \theta}$$

$$\overset{\text{tr.}}{=} \sum_{t=1}^{T} \frac{\partial \mathcal{L}_t}{\partial z_t} \sum_{k=\tilde{t}}^{t} \frac{\partial z_t}{\partial z_k} \frac{\partial^+ z_k}{\partial \theta}. \tag{97}$$