# OpenReview forum: "On the difficulty of learning chaotic dynamics with RNNs"
_NeurIPS.cc/2022/Conference — NeurIPS 2022 Accept_

### Official Review · Reviewer_Dedw · 2022-06-18

**Rating:** 7
**Confidence:** 4
**Soundness:** 3 good
**Presentation:** 4 excellent
**Contribution:** 3 good

**Summary:**

This paper concerns the issue of using RNNs to learn chaotic dynamics, a phenomenon exhibited in many important applications. To address the exploding/vanishing gradient problem (EVGP) when training RNNs, other works have typically proposed specific RNN architectures, however, these architectural choices rule out chaotic dynamics. This work performs theoretical analysis to formalize these ideas by relating the loss gradients of RNNs to the RNN dynamics using Lyapunov exponents. A result is presented for RNNs that converge to a stable fixed point or cycle showing that the loss gradients of these models will not diverge. A result for RNNs converging to chaotic dynamics shows that these models' loss gradients will inevitably diverge. A result for quasi-periodicity shows that there exist cases for bounded non-chaotic RNNs whose loss gradients can diverge.

Given these theoretical results, the work then performs an empirical evaluation to show the practical implications of the theoretical results. The main idea is that since we cannot both control EVGP with architectural choices and learn chaotic dynamics, we need to use a different approach to control EVGP. This paper suggests the use of a form of teacher forcing where at sparse time points the "true" state is computed given the observation and weights at that timestep. This has the effect of truncating the loss gradients.  The particular sparse time points are determined by the system's Lyapunov spectrum using the predictability time. Examples are provided using a Lorenz system, Duffing oscillator and weather data, showing that this approach performs better than backpropagation through time BPTT with gradient clipping as well as truncated BPTT where the initial states are set to zero and the truncated timepoints are chosen using the predictability time.

**Questions:**

1. If I am not mistaken, it seems to me that "sparsely forced BPTT" is basically a form of truncated BPTT where the truncation points are determined by the Lyapunov exponents, and the initial state that kicks off the truncated segment is the computed "true" state (using the TF approach) from the last step of the previous segment. It seems it would be helpful to the reader to point this different viewpoint out and perhaps highlight any practical differences. Section A.6.4 starts to make this connection but it seems a more thorough discussion in the main text would be helpful to the reader. Please let me know if I am misunderstanding something though.

2. The baselines used for the empirical results are: BPTT with gradient clipping and truncated BPTT where the initial state of each truncated segment is set to zero and the truncation points are determined by the Lyapunov exponents. This latter baseline shows that just truncating the segments by  $\tau_{pred}$ is not sufficient and using the teacher forcing approach is important. But this raises the question of the necessity of using $\tau_{pred}$ for the windowing? I.e., how would this approach do using evenly (or randomly) spaced truncation points but using the TF method to initialize the initial state? It is currently difficult for the reader to judge if using $\tau_{pred}$ is actually important. Another baseline, (though I would rank this as less important than the previously mentioned one), might be to learn the initial state as opposed to setting it to zero, similar to the "multiple shooting" ideas mentioned in the related work. I think these additional baselines, particularly the first one, would help strengthen the claims related to the proposed training method.

3. How practical is computing the predictability time, $\tau_{pred}$, for general systems? It is noted that the maximal Lyapunov exponents for the Lorenz and Duffing systems were taken from the literature and this exponent was computed using the TISEAN package for the weather data. It seems a discussion of computing this exponent is warranted to give the reader a better sense of what is required to get this value. The Vogt et al. 2022 paper that is cited seems to propose a method to estimate the Lyapunov spectrum. Perhaps the usefulness or limitations of that method could be discussed?



**Limitations:**

The authors acknowledge they have not proposed a full solution to the main problem, but rather have helped to formalize the problem and offer potential directions.

**Strengths And Weaknesses:**

Originality: While other works (cited in the related work section) have explored the connections between Lyapunov exponents and loss gradients before, this appears to be the first work to present formal theory for this connection in general RNN architectures, particularly for the chaotic case.  While the "sparsely forced BPTT" appears to be a specific variant of truncated BPTT (though see Question #1 related to this below), the teacher forcing approach seems useful and the idea to base the truncation points on the Lyapunov spectrum appears to be novel.

Quality:  The theoretical results appear to be sound (though I did not go as closely through every line of the additional proofs in the appendix). While the authors acknowledge they have not fully solved the difficult problem of controlling EVGP for problems with chaotic dynamics, the theoretical results point to attempting to address this with training methods as opposed to architectural constraints and the empirical results help to illustrate the proposed training method.  However, it seems some lingering questions related to the effectiveness of using the predictability time to choose the truncation points could be resolved with a few additional baselines (see Question #2 below). It is also currently unclear how practical using this predictability time is for more general problems (See question #3 below)?

Clarity: The paper is well-written and clear.

Significance: I think this paper is useful in formalizing the problem of attempting to control the EVGP while also attempting to learn chaotic dynamics with RNNs. Given the prevalence of chaotic systems, this is an important problem. The point that this problem cannot be addressed through architectural constraints is important and can hopefully lead to future training methods that can successfully address this.

---

> ### Author Response · Authors · 2022-08-02
> **Response to Weaknesses and Questions**
>
> We thank the referee for her/his generally positive assessment, for appreciating the importance and novelty of our approach, and for providing further helpful directions!
>
> *Questions:*
>
> *Q1 (truncated BPTT)*: Yes, one may see our approach as a form of truncated BPTT, but with the all-important differences that 1) we suggest a theoretically informed choice of the optimal ‘truncation length’ (forcing interval) and 2) a systematic procedure is provided for replacing current latent states by control values obtained by inversion of the observation model. Both these ingredients are indeed crucial, as shown in sects. 4.2 \& 4.3 (as well as Appx. A.6): Performance peaks around $\tau_{pred}$ and degrades for smaller or larger intervals (as illustrated in Fig. 2 and shown more systematically in Figs. 1 \& 3 and Appx. A.6). Likewise, just truncating the gradient is not sufficient, even when performed at optimal intervals ($\tau_{pred}$), since from a dynamical point of view we also need to re-calibrate the diverging latent trajectory.
>
> As suggested, we will make these points more explicit in the main text in sect. 4.1 by adding:
>
> “Note that one may see sparsely forced BPTT as a kind of truncated BPTT, but with the all-important differences that we 1) suggest a theoretically informed choice of the optimal ‘truncation length’ (forcing interval) and 2) provide a systematic procedure for replacing current latent states by control values obtained by inversion of the observation model. As shown in the next sections, both these aspects are crucial to avoid diverging gradients and keep model-generated trajectories on track whilst not loosing relevant longer time scales.”
>
> Since we are currently still bound by the 9-page limit for the main text, we will add this to the final revision with extended page-limit.
>
> *Q2 (additional baselines)*: The referee asks – if we understood correctly – whether using $\tau_{pred}$ is actually important, or whether TF by itself would do the job, using just any forcing (truncation) interval. Please note that this question is already answered in Figs. 1, 3, 5, 7, 8, 11, as these graphs systematically quantify the dependence of reconstruction quality (as measured by $D_{stsp}$ and $D_H$) on the interval $\tau$ used for TF: The minima around $\tau_{pred}$ indicate that choosing the theoretically informed values indeed works better than using other (smaller or larger) ‘window lengths’ (as also illustrated in Fig. 2). We would like to ask the referee to please get back to us should we have misunderstood her/his point!
>
> Nevertheless, we also took up the suggestion to check random intervals (new Appx. Fig. 14). As expected, if we choose the mean around $\tau_{pred}$ but increase the variance around this mean, performance degrades.
>
> Learning the initial states (rather than deducing them directly from the data) is certainly also a possibility when combined with $\tau_{pred}$, but would constitute a less direct strategy than the one we have chosen, requiring additional parameters to be trained.
> However, the referee’s remark alerted us to another potentially important control that would challenge our assumptions: Using truncated BPTT with the initial state forwarded from the last iteration, i.e. $z_{1(k+1)} = F(z_{\tau(k)})$ where $k$ indexes the window, rather than resetting it to zero! We included this additional control as new Appx. Fig. 9(c); as shown, just forwarding the states while truncating the gradients performs at least as bad as windowing with zero resetting.
>
> Other additional new controls are collected in Appx. A.6.4 \& A.6.6 in Figs. 9(b) \& 12.

---

> > ### Author Response · Authors · 2022-08-02
> > **Response to Q3**
> >
> > *Q3 (computing predictability time)*: This is indeed an important point, and we agree it needs more discussion. We computed the Lyapunov exponents using TISEAN for both the weather data (Figs. 3, 4) as well as the electrophysiological (EEG) data (Figs. 10, 11). Both are fairly complex and rich empirical data sets, so it’s certainly doable! Besides TISEAN, by now there are many other packages (e.g. DynamicalSystems.jl in Julia) available for this purpose (the method in Vogt et al., 2022, however, is not suitable here since we need something that works directly on the data). Computing maximal Lyapunov exponents from data thus became a fairly common procedure in recent years and takes little time (some domain knowledge for the data at hand may be required for appropriate filtering and de-trending, but this is not specific to the computation of Lyapunov exponents, of course, and should be done in general).
> >
> > We added a pg. at the bottom of sect. 4.1 (p. 7) on this issue, and will further expand on this in our main text Discussion once we are provided an additional page for the revision. We will add:
> >
> > “As noted in sect. 4.1, fairly standard packages are available for computing maximal Lyapunov exponents from data. Some background knowledge, as provided in classical textbooks (e.g. Ch. 5 in [43]), may be required for properly reading the output from these packages: Essentially, one would be looking be for a linear scaling region as in Figs. 3a \& 11a, ignoring both the initial noise transient as well as the plateau caused by reaching the full attractor extent. If unsure about the exact value, a moderate amount of jittering around the estimated mean value (see Appx. Fig. 14) may help.”

---

> > > ### Comment · Reviewer_Dedw · 2022-08-04
> > > **Thank you for your detailed reply**
> > >
> > > Thank you for expanding the discussion in relation to the similarities and differences between this approach and TBPTT. I think this will help the reader have a better understanding of the differences.
> > >
> > > I believe I slightly misunderstood the Figures in my initial read (I think related to reviewer 94xf comments on the presentation of the figures) and agree these figures address my question regarding the evenly spaced baseline I suggested. I also appreciate you including the randomly spaced baseline in the appendix as well as the additional baselines.
> > >
> > > I think the expanded discussion of computing the predictability time is helpful.
> > >
> > > My main concerns have been addressed and I have revised my score upward.

---

> > > > ### Author Response · Authors · 2022-08-06
> > > > **Thanks!**
> > > >
> > > > We thank the referee for the kind reply and time taken to evaluate our manuscript and rebuttal. We are glad we could address your concerns, and agree that the suggested comparisons and discussion are very helpful!

---

### Official Review · Reviewer_3EGL · 2022-07-10

**Rating:** 9
**Confidence:** 4
**Soundness:** 4 excellent
**Presentation:** 4 excellent
**Contribution:** 4 excellent

**Summary:**

The hidden states of recurrent neural networks are analyzed for both chaotic and non-chaotic time series and proved to suffer from exploding gradients in case of approximating the former type of dynamics. Furthermore, an efficient method to still train RNNs on chaotic time series is proposed and exhaustively validated on synthetic and real-world data.

**Questions:**

- In line 121 an explanation of $h$ could not be found. How does the author's formulation relate to the conventional $\mathbf{z_t} = f((\mathbf{W_s}\mathbf{s_t} + \mathbf{B_s}\mathbf{s_t}) + (\mathbf{W_z}\mathbf{z_{t-1}} + \mathbf{B_z}\mathbf{z_{t-1}}))$ notation, where two different weight matrices $\mathbf{W_s}$ and $\mathbf{W_z}$ are used for the input and past hidden state mapping, respectively?
- In equation (17) the authors suggest to choose the teacher forcing frequency of the hidden state as $\tau_{\text{pred}} = \frac{\ln2}{\lambda_{\text{max}}}$. Hence, it depends on the largest Lyapunov exponent $\lambda_{\text{max}}$ (which likely changes during training). Does this mean, $\tau_{\text{pred}}$ should be assessed and updated regularly during training, based on a repeatedly computed $\lambda_{\text{max}}$? I might be wrong with my assumption of a changing $\lambda_{\text{max}}$ during training, when it is calculated on the training data and not on $\mathbf{z_t}$ (I understood it to be the latter, though). A brief explanation of how $\lambda_{\text{max}}$ was computed (using the training data or the RNN's hidden state) would be highly appreciated.
- Figure 1 indicates that LSTMs behave more consistent with the proposed theory than RNNs. Did the RNNs have more difficulties in learning the processes in general, which shows up in the $D_{stsp}$ and $D_{H}$ analysis, or how do the authors explain this deviation between RNNs and LSTMs? A plot like Figure 2 for RNNs would be appealing and informative.

**Limitations:**

Limitations are addressed by the authors.

**Strengths And Weaknesses:**

### Originality
Strengths:
- Computing the largest Lyapunov exponent $\lambda_{\text{max}}$ of the RNN orbit appears to be an appealing and highly informative measure about the RNN's nature.

### Quality:
Strengths:
- The problem of modeling chaotic time series and the proposed analysis are of high value for the ML community.
- Theoretical proofs are excellently verified empirically on a broad variety of synthetic and real-world processes with varying difficulty, suggesting a fairly simple but efficient method of how to train RNNs on chaotic time series.

Weaknesses:
- The authors might include [this work](https://journals.aps.org/prl/abstract/10.1103/PhysRevLett.120.024102), where an ESN has been shown to successfully predict chaos.

### Clarity and significance:
Strengths:
- According to my assessment, the analysis of exploding gradients when trianing an RNN on chaotic time series is of very high relevance for the ML community and thus constitutes a high significance of this work.
- The relevance and significance of this work is further amplified by the proposed effective method to train RNNs on chaotic time series.
- Despite the complex nature of the discussed problem, the paper is written and structured particularly clear and applicable.

Weaknesses:
- In line 172, does $\rho$ relate to the spectral radius? Might be helpful for the readers to specify.
- In line 173, please add "equation" before (1) to a facilitate application.

---

> ### Author Response · Authors · 2022-08-02
> **Response to Weaknesses and Questions**
>
> We are very glad to hear the referee found our work highly relevant and significant, and the empirical evaluation excellent!
>
> We addressed all minor weaknesses as suggested. In particular, we included and discussed the cited work in sect. 2, thank you for pointing this paper out to us, indeed relevant \& interesting! We also addressed the issues on lines 172 \& 173 (lines 175 \& 176 in rebuttal version, see blue highlighting in updated manuscript).
>
> *Questions:*
>
> Q1: In eqn. (1) we used a very general formulation of RNNs as recursive maps to allow for common mathematical treatment ($F$ denotes this generic map, while $f$ is the specific activation function used). But this is really just mathematical notation, the specific RNN formulations used later in the specific parts of the proofs (sect. A.2) and in the empirical evaluation (sect. 4) were indeed conventional RNNs, with different weight matrices for past hidden states and inputs, and the very same form as most commonly employed in the literature (see sect. A.1.4 for definition of LSTMs and A.1.3 for PLRNNs, for instance). We will add this information and make this point more explicit in the main text, and also add the explanation of $h$ (the bias term), once we are given the additional page for implementing the revisions (right now the constraint is still on 9 pages).
>
> Q2: Thanks for pointing out this confusing aspect in our presentation, this is an important hint! We now added a more detailed explanation on how these exponents were obtained at the bottom of sect. 4.1, p. 7. They were indeed obtained from the *empirical* time series on which any of the RNNs is to be trained, not from the RNNs themselves (they are, however, fairly easy to compute also for RNNs based on their known Jacobians, see e.g. sect. A.2). This is because the Lyapunov spectrum is an invariant characteristic of the *empirical system* (or time series data) we try to reconstruct (approximate) through RNN training, i.e. well-trained RNNs must exhibit the same Lyapunov exponents in the end if they are to reproduce the observed time series data. The maximal Lyapunov exponent therefore needs to be computed only *once* on the true data, for which there are many out-of-the-box and fast packages around (e.g., Julia: DynamicalSystems.jl, C++: TISEAN).
>
> Q3: Yes, the referee guessed correctly, vanilla RNNs are indeed somewhat harder to train on these data than LSTMs (presumably cos they struggle more with longer-term dependencies in the data). A plot similar to Fig. 2 for vanilla RNNs is now included as Appx. Fig. 13 (but please note that this is just a single example of course, it does not reflect the differences in ease of training that one sees in the group statistics in Figs. 1 \& 3!).

---

### Official Review · Reviewer_94xf · 2022-07-12

**Rating:** 7
**Confidence:** 3
**Ethics Flag:** Yes
**Soundness:** 3 good
**Presentation:** 3 good
**Contribution:** 4 excellent

**Summary:**

This paper studies the problem of training RNNs to learn chaotic dynamics. First, the paper draws analytic connections between exploding and vanishing gradients (due to the repeated product of Jacobian terms when computing gradients), and dynamical systems characterized by Lyapunov exponents. The paper proves that in order for the gradients and loss of an RNN to be well behaved, the RNN must converge to a limit cycle or fixed point. The consequence of this is that it will be difficult to use gradient-based optimization to train an RNN to learn chaotic dynamics.

The paper proposes a new solution to address this problem: the idea of resetting network dynamics periodically to a "best-fit" state given the readout weights. They call this method "sparsely forced" BPTT, and compare it to other methods of stabilizing RNN training (gradient clipping, and windowing the time series into distinct sections). The paper also provides a heuristic for selecting how often to force/reset the state, based on the maximum Lyapunov exponent of the underlying DS.

**Questions:**

Rather than use a fixed learning interval (tau), would there be any benefit to annealing tau over time? A natural choice in my mind would be to start small and slowly grow tau, and use early stopping with a validation set to determine when the network has started to diverge? Curious if the authors think this is reasonable.

**Limitations:**

yes

**Strengths And Weaknesses:**

Strengths
1. The paper is well written, clearly motivated, and has an appropriate amount of context (with additional detail present in the supplement). I found it a pleasure to read.
2. The problem the paper addresses is significant and, to my knowledge, their proposed solution is novel.
3. I really like the simplicity of sparsely forced BPTT.

Weaknesses
1. For the empirical evaluation, the authors mention they compare against a baseline of BPTT with gradient clipping as well as with a simple windowing scheme (resetting the state to zero). I think these comparisons should be more fleshed out in the main text. For gradient clipping, what values of gradient clipping were tried? Was it clipping by value or shrinking the norm? For windowing, what window lengths were tried? Does the sparsely forced BPTT outperform all possible choices for those hyperparameters?

Minor suggestions
- I don't have a concrete suggestion here, but I feel like Figures 1 and 2 emphasize the wrong information. As drawn, they emphasize the difference between the three architectures (that's what the eye gets drawn to first), but the important comparison is between the solid lines (the proposed method) and the dashed lines (the baseline method). I wonder if there is someway of regenerating those figures that puts the emphasis on the comparison between ST-BPTT and gradient clipping.
- Line 310: Instead of saying that the value of the exponent is in close agreement with previous work, why not just say what the value is from that previous work?
- I wonder if there is a simple figure or toy example that can demonstrate Theorem 1 and 2. For example, if one were to train a toy RNN on a chaotic system, can you show how the gradient terms diverge?

---

> ### Author Response · Authors · 2022-08-02
> **Response to weakness, minor suggestions, and question**
>
> We are happy to hear the referee liked our paper and appreciated the significance of our theoretical treatment and empirical results!
>
> *Weakness (hyper-parameter settings):*
>
> We had tried a couple of different hyper-parameters, but as suggested a more systematic evaluation of the role of different clipping procedures (Euclidean vs. max norm clipping) for different clipping thresholds and of different windowing lengths is now provided in new Appx. Fig. 12 and new Appx. Fig. 9(b), respectively, and referred to in the main text (in sect. 4.2 below Fig. 2, p. 8). Other additional controls with a different initialization procedure and random window lengths can be found in new Fig. 9(c) and new Fig. 14. But yes, sparsely forced BPTT outperforms all of these choices.
>
> *Minor points:*
>
> 1) We can see the referee’s point, and thought about how to best modify the figures. One possibility would be to separate the graphs for the different RNNs into different panels. But then possibly the point is lost that our findings are *general* across RNN architectures (the minimum always occurs around $\tau_{pred}$), plus we’d need much more space for the figures. So we decided to simply  “de-accentuate” the differences among RNNs by using different shades of the same color for the lines, rather than three different colors. We hope the referee agrees with this solution (see updated manuscript).
> 2) Line 310 (now line 319 in rebuttal version): Done!
> 3) We added a simple illustration of the behavior of the loss gradients for limit cycle and chaotic regimes as Appx. Fig. 15 (if space permits, we will move this to the main text later).
>
> *Question (annealing procedure):*
>
> Thanks for bringing this up! Yes, this is indeed a very reasonable and good idea. Starting with short forcing intervals initially to first get the short-term behavior right, and then challenging the system more and more for longer time spans until the Lyapunov horizon is reached, may be a good strategy and further improvement. We actually gave it a quick try, but our impression is it would need a much more thorough evaluation in its own right (at least a ‘naive’ implementation as indicated above did not seem to further boost performance; we could add a figure on this if deemed useful). We will therefore rather leave this idea as an outlook in our Discussion (once the main text page limit is lifted) as follows:
>
> “A further interesting direction for improvement might be to regulate the forcing interval $\tau$ through an annealing procedure, for instance starting at $\tau=1$ and ramping up to $\tau=\tau_{pred}$ throughout training. The idea here would be to first get the short-term behavior right, and then challenge the system more and more for longer time spans until the predictability time is reached.”
>
> We will appreciate the suggestion by the referee in our Acknowledgments.

---

> > ### Comment · Reviewer_94xf · 2022-08-06
> > **Thanks for your response**
> >
> > Thanks for your response!

---

### Official Review · Reviewer_okuc · 2022-07-12

**Rating:** 7
**Confidence:** 2
**Soundness:** 3 good
**Presentation:** 3 good
**Contribution:** 3 good

**Summary:**

This paper demonstrates that the loss gradients are inherently linked to RNN dynamics. When RNN dynamics converge to a fixed point or cycle, the paper shows that loss gradients will remain bounded. When RNN dynamics are chaotic, gradients will always explode and cannot be addressed by architectural designs or gradient clipping. The paper proposes one way of addressing the problem on chaotic data, by tracking the Lyapunov spectrum during training and uses teacher forcing when local divergence rates becomes too large.

**Questions:**

- It is unclear to me why other forms of constraints and regularization cannot be used to alleviate the exploding gradient problem, which the sparsely forced BPTT can. I'd appreciate a more intuitive explanation from the authors.

**Limitations:**

Yes

**Strengths And Weaknesses:**

### Strengths:
- This paper is decently well written and has a good overview of related works.
- The paper shows that the exploding and vanishing gradient problem in RNNs can be easily addressed when the RNN is well-behaved, but will necessarily be problematic when modeling chaotic dynamics with RNNs. This principled view into the problem and the conditions on which they occur appear to be significant.
- The paper proposes sparsely forced BPTT as a way to enable the learning of chaotic dynamics.

### Weaknesses
- It's not clear how applicable the sparsely forced BPTT method would be in real-world applications, as it would be expensive to calculate the Lyapunov spectrum throughout training.

---

> ### Author Response · Authors · 2022-08-02
> **Response to Weakness and Question**
>
> We thank the referee for her/his generally positive evaluation and important remarks that led us to clarify and expand on these points in the paper.
>
> *Weakness (applicability to real-world datasets \& computation of Lyapunov exponents):*
>
> First, please note that we used two fairly complex and rich real-world datasets in our evaluation: Climate data (Figs. 3, 4) and electrophysiological (EEG) data (Figs. 10, 11), for both of which we computed Lyapunov exponents.
>
> However, there also seems to be a misunderstanding here (thanks for alerting us!): We only need the maximal Lyapunov exponent (not the whole spectrum), and – more importantly – we only need to compute it *once* *before* training (not during training)! This is because the Lyapunov spectrum is an invariant characteristic of the *underlying true system* (conveyed through the data) that the trained RNN must inherit for a successful reconstruction. For this purpose there are many out-of-the-box and fast packages around (e.g., Julia: DynamicalSystems.jl, C++: TISEAN). Hence, determining this exponent is quick and ‘cheap’ compared to the actual training.
>
> We now clarified this at the end of sect. 4.1 on p. 7. We will also come back to this important point in our Discussion once the page limit is lifted.
>
>
> *Question (intuitive explanation why constraints don't work):*
>
> First, we are not ruling out in general that someone may be able to come up with a smart regularization scheme which could fix this issue. However, our analysis shows that this will not be easy as any such scheme would need to fulfill a number of requirements. This is not the case for any of the regularization schemes or constraints suggested so far, as they either a) do not satisfactorily address the exploding gradient problem on chaotic time series, or b) are provenly (see sect. A.1.6 and sect. 2) too restrictive to allow for generation of chaotic behavior in the trained RNN.
>
> The intuition here is the following: As proven in sect. 3.3 (Theorem 2), chaotic systems will inevitably cause diverging gradients. Hence, to avoid exploding gradients while training RNNs on time series from chaotic systems, one either needs to constrain the RNN so much that chaotic behavior is completely disabled to begin with (i.e., ultimately by forcing all Lyapunov exponents to be smaller or equal to zero), implying a very poor fit to such data. Or one needs to be a bit more lenient and thereby allow for the possibility of exploding gradients (as LSTMs or PLRNNs in fact do). Thus, our suggestion is to put the focus more on the training process itself and address the problem at that level, a possibility we think that has not been so widely appreciated yet.
>
> We will add this kind of intuitive explanation to sect. 5 (Discussion) once we are given an additional page for the revisions (right now the constraint is still 9 pages).
>
> Please also note that we added a number of new experiments and controls now in Appx. A.6.4 and A.6.6.

---

### Author Response · Authors · 2022-08-02
**Thanks for constructive and supportive comments!**

We thank all four referees for their very constructive, positive and supportive feedback! We are happy to hear that all referees found our work important and significant for the field.

We addressed all comments through a number of further experiments (collected in Appx. A.6.4 \& A.6.6), new figures (7 new graphs), and appropriate textual changes as outlined in our point-by-point replies below. Textual changes were highlighted in blue in the uploaded rebuttal version (please note, however, that we could not yet accommodate all intended changes in the main text cos of the current page restrictions which will only be lifted after the discussion period).

---

### Meta-Review · Area_Chair_EvEC · 2022-08-28

**Recommendation:** Accept
**Confidence:** Certain

**Metareview:**

This paper studies the ability for an RNN to learn the dynamics of a chaotic system. The authors relate the learning dynamics for these systems to the Lyapunov exponents of the underlying dynamical system. They then propose ameliorating the role of chaotic dynamics by “sparsely forced BPTT” which forces the RNN hidden state on a timescale that is induced by the Lyapunov exponents.

All of the reviewers supported accepting this paper and it was the highest rated paper on my stack. Reviewers cited the clear writing and motivation as well as the simplicity and effectiveness of the proposed solution. Reviewers also noted the importance of time-series analysis as a research problem. Improving modeling of chaotic time series seems like a valuable and important contribution.

**Award:**

Yes

---

### Decision · Program_Chairs · 2022-09-14

Accept